# Hetero-oligomerization of TDP-43 carboxy-terminal fragments with cellular proteins contributes to proteotoxicity
Akira Kitamura [1,2,3,5] ✉, Ai Fujimoto [1,5], Rei Kawashima[3], Yidan Lyu[3], Kotetsu Sasaki[1], Yuta Hamada [1], Kanami Moriya[3], Ayumi Kurata[3], Kazuho Takahashi[3], Reneé Brielmann[4], Laura C. Bott [4], Richard I. Morimoto[4] & Masataka Kinjo[3]

Carboxy terminal fragments (CTFs) of TDP-43 contain an intrinsically disordered region (IDR) and form cytoplasmic condensates containing amyloid fibrils. Such condensates are toxic and associated with pathogenicity in amyotrophic lateral sclerosis. However, the molecular details of how the domain of TDP-43 CTFs leads to condensation and cytotoxicity remain elusive. Here, we show that truncated RNA/DNA-recognition motif (RRM) at the N-terminus of TDP-43 CTFs leads to the structural transition of the IDR, whereas the IDR itself of TDP-43 CTFs is difficult to assemble even if they are proximate intermolecularly. Hetero-oligomers of TDP-43 CTFs that have recruited other proteins are more toxic than homo-oligomers, implicating loss-of-function of the endogenous proteins by such oligomers is associated with cytotoxicity. Furthermore, such toxicity of TDP-43 CTFs was cell-nonautonomously affected in the nematodes. Therefore, misfolding and oligomeric characteristics of the truncated RRM at the N-terminus of TDP-43 CTFs define their condensation properties and toxicity.

Protein aggregation is a principal phenomenon in which misfolded proteins and intrinsically disordered proteins (IDPs) assemble under certain conditions. Membraneless intracellular assemblies in subcellular compartments of cells and tissues are called protein aggregates or condensates, which are formed by liquid-liquid or liquid-solid phase separation, respectively, and have the ability to concentrate biopolymers[1]. Although various IDPs contain intrinsically disordered regions (IDRs), IDRs are known to play functional roles that diverge from those of folded proteins/domains and revolve around their characteristics to act as hubs for protein-protein interactions because IDR conformations are dynamically and rapidly interconverting between free and bound states[2,3]. Furthermore, aggregation propensities are not involved in IDRs but in domains of protein except IDRs[4]. However, protein misfolding is known to occur in various situations co- and post-translationally as a result of cell state transitions during stress[5,6]. Misfolded proteins are recognized by various cellular mechanisms for protein quality control and can be either refolded by molecular chaperones, or targeted for degradation by the ubiquitin-proteasome system and the autophagy-lysosomal pathways to maintain protein homeostasis (proteostasis) in cells[5–7]. If protein quality control fails, misfolded proteins accumulate as aggregates or condensates in vivo and in vitro. Therefore, it is thought that

protein misfolding and conformational transition of IDRs are independent physicochemical properties. Biomolecular condensates including protein and other biomolecules such as RNA are defined as membraneless organelles, which have specialized functions in cells; also, it is then summarized as a new name for cellular inclusion bodies, aggregates, membraneless organelles, and puncta[8]. Condensates also form as a physiological process, for example during cell stress, or in development, even though they are not diseases; however, they are also frequently found in the cytoplasm and nucleus in neurons, muscles, etc., when in neurodegenerative diseases and a certain myopathy[9,10]. Since one of the cardinal features of neurodegenerative diseases is known to be the presence of protein toxicity (also called proteotoxicity), it is suspected that the toxicity of aggregation-prone proteins that form condensates is associated with neurodegenerative diseases such as amyotrophic lateral sclerosis (ALS), frontotemporal lobar degeneration (FTLD), Alzheimer's disease, and Parkinson's disease[11,12]. Such disease-associated condensates are deposited inside and outside the cell, and mature and deposited protein aggregates of such disease-associated proteins (e.g., α-synuclein, Aβ, polyglutamine proteins, etc.), called amyloids, are initially thought to be a proteotoxic cause; however, in recent years, their soluble aggregates, the smaller of which are usually called oligomers and

[1]Laboratory of Cellular and Molecular Sciences, Faculty of Advanced Life Science, Hokkaido University, N21W11, Kita-ku, Sapporo 001-0021, Japan. [2]PRIME, Japan Agency for Medical Research and Development, Chiyoda-ku, Tokyo 100-0004, Japan. [3]Laboratory of Molecular Cell Dynamics, Faculty of Advanced Life Science, Hokkaido University, N21W11, Kita-ku, Sapporo 001-0021, Japan. [4]Department of Molecular Bioscience, Rice Institute for Biomedical Research, Northwestern University, Evanston, IL 60208, USA. [5]These authors contributed equally: Akira Kitamura, Ai Fujimoto. ✉e-mail: akita@sci.hokudai.ac.jp

protofibrils, have been reported to be highly proteotoxic species[13–15]. Thus, how oligomers are formed and whether they are toxic are important questions.

More than 97% of both sporadic and familial ALS, transactive response element (TAR) DNA/RNA-binding protein 43 kDa (TDP-43)-positive condensates in the cytoplasm of the affected neurons are featured[16]. TDP-43 condensates are also the hallmark of some forms of frontotemporal dementia (FTD). Such FTDs are classified as FTLD with TDP-43 pathology (FTLD-TDP)[17]. FTLD-TDP subtypes (typically, type A–D) are likely to be distinguished by the type of insoluble proteins in addition to TDP-43[18]. In ALS and FTLD pathology, not only full-length TDP-43 but also TDP-43 CTFs are included in the condensates in the brain but are rarely detected in the spinal cord[19,20]. Therefore, it has been proposed that TDP-43 CTFs may not be a prerequisite for neurodegeneration[21,22]; however, TDP-43 fragmentation and its CTFs are a potential therapeutic target because ectopic expression of TDP-43 CTFs leads to various dysfunctions such as impaired autophagy and proteasome function, reducing functional dimerization/oligomerization of TDP-43, reduced motor function, and cell death, in cellular models and also in animal models[23–27]. Consequently, the proteotoxic mechanism of TDP-43 CTFs remains elusive.

Human TDP-43 (hTDP-43) carries an N-terminal ubiquitin-like domain, two RNA/DNA-recognition motifs (RRM1 and RRM2; hTDP-43 106–176 and 191–262, respectively), and a C-terminal glycine-rich region (GRR; hTDP-43 274–414) that is known to be a prion-like domain[28], including a glycine-rich (hTDP-43 274–310), a hydrophobic patch (318–343), the Q/N-rich (344–360), and glycine/serine-rich domains (hTDP-43 368–414), in addition to several amyloidogenic core regions[29,30]. The GRR is also known to be an IDR, which regulates interactions with proteins in other RNA-binding proteins (e.g., hnRNP A1/A2 and FUS/TLS) as well as the self-interaction of TDP-43 via liquid-liquid phase transition[31,32]. hTDP-43 contains typical DEVD-like motifs in regions 86–89 and 216–219 in the primary sequence, which can be cleaved by caspase 3 (DETD and DVMD sites, respectively). hTDP-43 CTFs, 90–414 (35 kDa) and 220–414 (25 kDa), are called TDP35 and TDP25, respectively[33]. Caspase 3 is not the case that produces TDP35 and TDP25, but calpain, alternative splicing, and protein clearance pathways are also responsible for producing such CTFs[22,34,35]. The assembly properties and characteristics of TDP25 condensates have been described in ref. 22. The TDP-43 condensates in the cytoplasm and the nucleus is associated with RNA granules[36]; however, the TDP25 condensates did not contain RNA[23]. Furthermore, the structures of the pathological filaments of the GRR were identified[37,38]; however, it remains unknown how TDP25 forms condensate in the cytoplasm. In the yeast model, although the cytoplasmic condensates increased according to how many regions of the RRM2 of hTDP-43 were included, the GRR itself did not form condensates despite the fact that region 311–320 is known to be the most critical for the amyloidogenic fibrils[39,40]. Although intact RRM2 is made up of two α-helixes (α1–α2) and five β-strands (β1–β5), β3 and β5 were reported to form non-amyloidogenic fibrils in vitro[41]. These reports implicate that the N-terminal conformational change in the truncated RRM2 (tRRM2) of TDP25 could lead to aggregation. Various pathological and responsible enzymatic TDP-43 cleavage sites that are potentially producible its CTFs have been identified (e.g., 208, 224, 219, 229, 243, 246, 247, etc.) but its cleavage mechanism is not still fully understood[26]. Therefore, analyzing the aggregation propensity and toxicity of artificially truncated TDP-43 CTFs involves a straightforward and recursive approach using mutants with systematically depleted amino acids.

To discover the pathogenic characteristics of TDP-43 CTFs, various disease models have been established using mice, rats, fruit flies, and worms in addition to tissue culture cells[22,26]. The nematode worm *Caenorhabditis elegans* has been instrumental to uncover tissue and cell dysfunction and dysregulation of longevity caused by misfolded proteins during especially aging because the lifespan is relatively short compared to mice and rats[42–44]. Pan-neuronal expression worm lines of yellow fluorescent protein (YFP)-tagged TDP25 showed significant movement defects associated with protein condensates in the neuronal cell bodies[45]. On the other hand, to demonstrate

proteotoxicity of misfolded proteins associated with human neurodegenerative disorders such as Aβ, expanded polyglutamine proteins, and superoxide dismutase 1 (SOD1), their expression in the body-wall muscle has been widely used[46–49]. Although studying the cell-nonautonomous effects of such misfolded proteins in neurons[49,50], there are few reports to demonstrate whether the proteotoxicity of misfolded proteins in the body-wall muscles affects the lifespan[46]. We reasoned that a new model that expresses TDP25 could be informative for elucidating proteotoxicity during ageing by misfolded protein oligomers and their condensates formed through a liquid-liquid phase transition. We introduced highly toxic TDP25 and further truncated TDP25, which is a highly oligomerized but low toxic variant in murine cells, into *C. elegans* body-wall muscle and analyzed the oligomerization status and protein mobility of TDP-43 variants in the condensates, as well as animal motility and lifespan.

Taken together, using murine neuroblastoma Neuro2a cells and *C. elegans*, we demonstrate the role of misfolding of tRRM2 and GRR as IDR of TDP-43 CTF during their condensation containing amyloidogenic aggregation by analyzing the efficiency of cytoplasmic condensation, molecular fluidity in condensates, and oligomeric states in lysate, of various TDP-43 CTFs. Furthermore, we investigated the proximity properties of the N- or C-terminal side of TDP-43 CTFs using a photo-induced oligomerization tag.

## Results

### The truncated RRM region at the N-terminus of TDP25 specifies the condensation efficiency

Although TDP25 (amino acid 220–414; hereafter called C220) is aggregation-prone[23], it remains elusive how C220/TDP25 aggregation is deposited in the cells. We first confirmed the aggregation propensity of C220, C263 (amino acid 263–414), and C274 (amino acid 274–414) tagged with a monomeric green fluorescent protein (GFP) at their N-terminus in murine neuroblastoma Neuro2a cells by observing distinct brighter structures than those are distributed in the cytoplasm (hereafter called condensates) using fluorescence confocal microscopy. C220 frequently formed cytoplasmic condensates, but both C263 and C274 (Consisting of GRR only) did not (Fig. 1a–c). Therefore, we hypothesize that the sequence of truncated RRM2 on the N-terminal side of the GRR specifies the aggregation and condensation propensity of the TDP-43 CTFs. Since aggregation analysis of TDP-43 CTFs with every 5 residues removed from the N-terminus of C220 up to C245 (amino acid 245–414) showed that they can be categorized into four distinct types[51], we investigate the condensate formation of GFP-tagged TDP43 CTF variants adding N-terminal 10, 20, and 30 amino acid residues to C263 (C233, C243, and C253, respectively; shown in Fig. 1a) in addition to C220 and C274 in Neuro2a cells was analyzed using fluorescence confocal microscopy and the solubility assay followed by western blotting of their cell lysates. C233, C243, and C253 formed condensates in the cytoplasm (Fig. 1b). However, the proportion of cells containing C233 condensates was extremely low, while that of C243 and C253 was higher than C233 and C263 but lower than C220 (Fig. 1c). C233, C243, and C253 were observed in the 1% SDS-insoluble pellet fraction, but their abundance in the pellet fraction was lower than that of C220 (Fig. 1d). For C220, ~80% was recovered in the insoluble fraction (Fig. 1d). For C233, ~40% was recovered in the insoluble fraction (Fig. 1d). For C263 and C274, ~20–30% was recovered in the insoluble fraction. It suggests that the sequence on the N-terminal side of GRR is responsible for the cytoplasmic condensation of TDP-43 CTFs. Furthermore, although the cytoplasmic condensates of GRR (C263 and C274) were rarely observed (Fig. 1b), they did not behave as soluble as monomeric GFP (Fig. 1c, d), suggesting they may form some complexes with cellular biomolecules. Next, as C220 and C233 contain a nuclear exporting signal (NES) candidate sequence, and C243 does partially but C253 does not, we confirmed whether abundance in nuclear and cytoplasmic localization bias could affect cytoplasmic condensations using C220 carrying mutated NES sequence (I239A, L243A, L248A, L249A, and I250A substitutions; G-C220mNES). The relative fluorescence intensity of G-C220mNES in the nucleoplasm to the cytoplasm

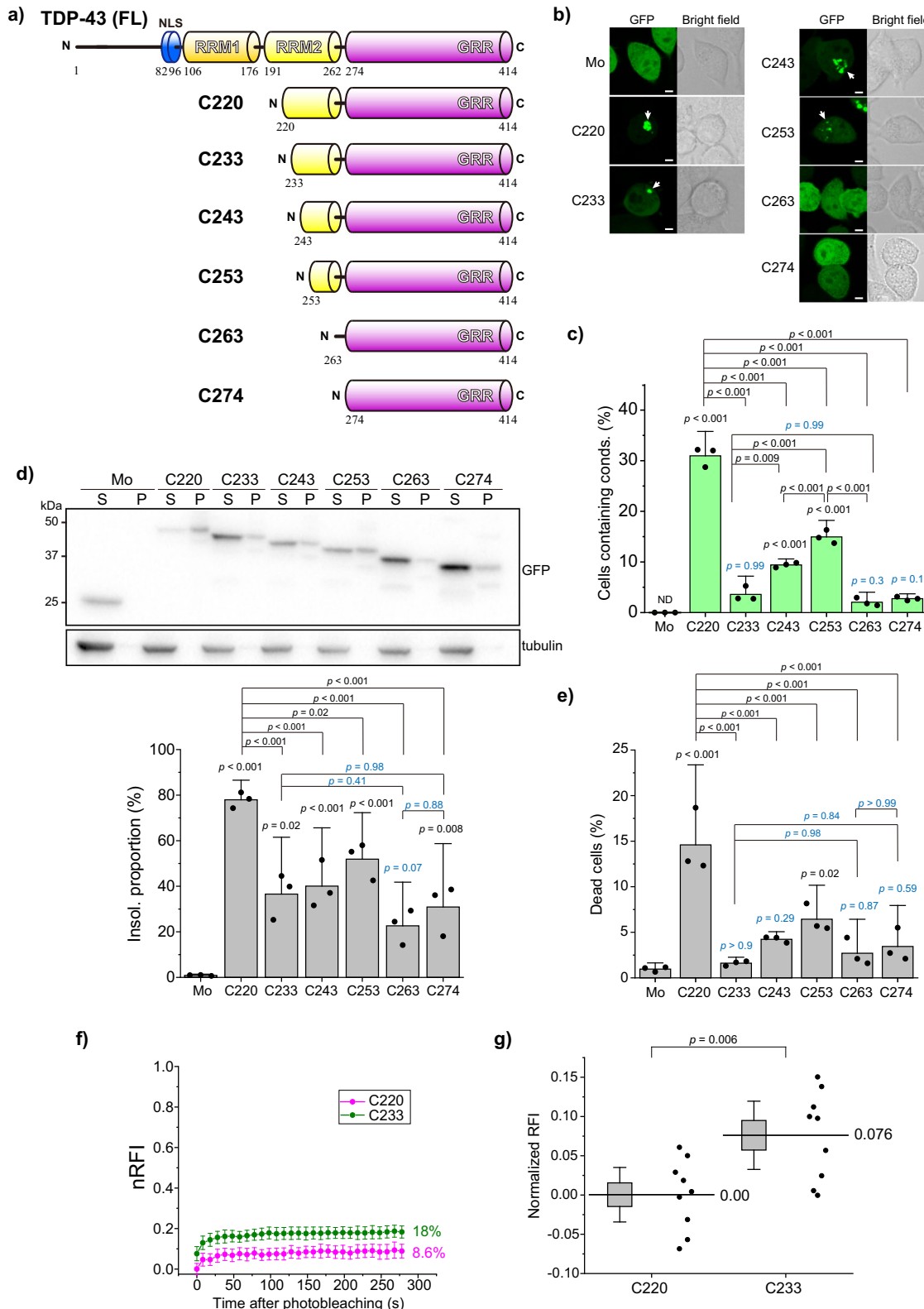

was not changed compared to that of G-C220 (Supplementary Fig. 1a and b). Cytoplasmic condensation efficiency of G-C220mNES was not also changed compared to that of G-C220 (Supplementary Fig. 1c). Accordingly, the condensation efficiency in these TDP-43 CTFs can be compared with or without NES sequence, suggesting that NES sequence in TDP-43 CTFs may not be active as reported in the full-length TDP-43[52,53]. The proportion of dead cells expressing TDP-43 CTFs showed a positively correlative relation

with that of cells containing condensates (Fig. 1c, e), suggesting that the condensates of TDP-43 CTFs are responsible for cytotoxicity. Cells expressing C233, C263, and C274 did not show an increase in cell death compared to the expression of GFP monomers as a negative control (Fig. 1e), suggesting that C233 and GRR are not cytotoxic.

To compare the mobility and fluidity of C220 and C233 in the cytoplasmic condensates, we performed fluorescence recovery after

**Fig. 1 | Formation of the cytoplasmic condensates of TDP-43 carboxy terminal fragments (CTFs) in Neuro2a cells. a** Primary structures and abbreviation of full-length TDP-43 (FL) and its carboxy-terminal fragments (CTFs): amino acids 220-414 (C220), 233-414 (C233), 243-414 (C243), 253-414 (C253), 263-414 (C263), and 274-414 (C274). The RRM and GRR denote the RNA/DNA-recognition motif and the glycine-rich region, respectively. **b** Confocal fluorescence and bright field microscopic images of Neuro2a cells expressing GFP monomers (Mo) and GFP-tagged CTFs. White arrows represent the cytoplasmic condensates. Bars = 5 µm. **c** The population of Neuro2a cells containing cytoplasmic condensates (conds). Bars: mean + 95% CI; Dots: raw values (*n* = 3 independent experiments). **d** Western blot stained using an anti-GFP and anti-α-tubulin antibody of cell lysates: 1% SDS-soluble (S) and insoluble (P) (top). The quantification of the abundance of TDP-43 CTFs in the insoluble fraction. The abundance shows the normalized band intensity in the P fraction to total (S + P) fraction (*n* = 3 independent experiments) (bottom). **e** The population of dead cells expressing TDP-43 CTFs tagged with GFP and GFP

monomers (Mo) using a propidium iodide exclusion test. Bars: mean and 95% CI (*n* = 3 independent experiments). **f** Comparison of time-dependent normalized relative fluorescence recoveries (nRFI) of GFP-tagged C220 and C233 (magenta and green, respectively) in live Neuro2a cells using fluorescence recovery after photo-bleaching (FRAP). Dots and bars indicate the mean and 95% CI (*n* = 9 independent cells). Inset values show mobile fraction from the maximum recovery of nRFI obtained by non-linear curve fitting analysis using a one-component exponential recovery model. The individual data points are available in Supplemental Data 1. **g** Comparison of nRFI just after photobleaching (*t* = 0) between C220 and C233. The highest and lowest points of the box represent SD, and its bars represent 95% CI. Dots represent raw values. Inset values show the mean. The raw FRAP data provided in Supplemental Data 1. In **c**, **e**, and **g**, *p*-values above the bars and lines: one-way ANOVA with Tukey's test compared to GFP monomers as control and comparison between lines, respectively.

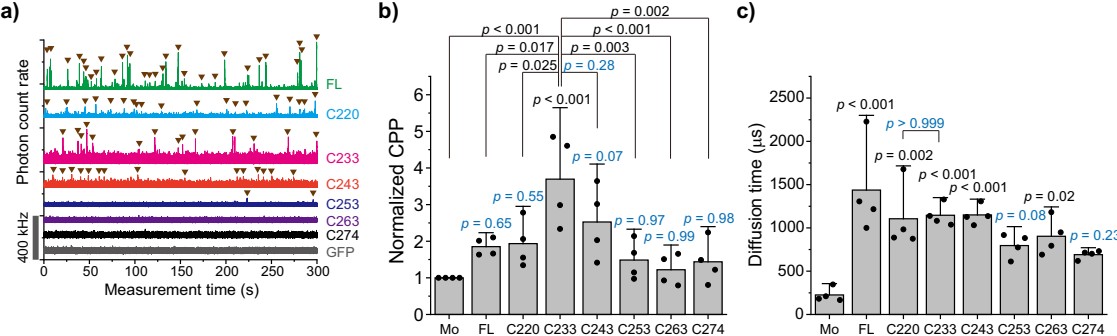

**Fig. 2 | Analysis of soluble oligomers of TDP-43 CTFs using fluorescence cor-relation spectroscopy (FCS). a** Typical photon count rate records during the FCS measurement of GFP monomers (Mo), full-length TDP-43 (FL), and CTFs in Neuro2a cell lysate. The Gray bar on the *y*-axis shows the range of 400 kHz of photon count rate. **b** The plot of normalized counts per particle (CPP), and relative fluor-escent brightness per single particle. Bars: mean + 95% CI; Dots: raw values (*n* = 4

independent experiments). **c** The weighted average of diffusion time was obtained by curve fitting analysis of the autocorrelation function using a two-component dif-fusion model. Bars: mean and 95% CI; Dots: raw values (*n* = 4 independent experiments). In **b** and **c**, *p*-values above the bars and lines: one-way ANOVA with Tukey's test compared to GFP monomers as control and comparison between the lines, respectively.

photobleaching (FRAP) in the TDP-43 CTF-expressing Neuro2a cells (Fig. 1f). Because the movement of fluorescent molecules during the pho-tobleaching time reduces the apparent photobleaching efficiency, the nor-malized relative fluorescence intensities (nRFI) that were corrected by the photobleaching efficiency of immobile GFP were analyzed. The nRFI of C233 just after photobleaching (at 0 s in Fig. 1f) was higher than those of C220 (Fig. 1g). The increase in nRFI of C233 just after the photobleaching suggests that a portion of C233 (7.6%; Fig. 1g) could be more rapidly associated and dissociated in the condensates than C220. However, there was no difference in the shape of the recovery curve, except for the nRFI value at 0s between C220 and C233 (Fig. 1f), indicating that the fluidity of C220 and C233 in the condensates may be the same.

Next, the oligomerization states of such GFP-tagged TDP-43 CTFs in the 1% Triton X-100-soluble fraction of the cell lysates were analyzed using fluorescence correlation spectroscopy (FCS), which can determine the homo-oligomerization of soluble and diffuse fluorescent species from the fluorescence fluctuation that occurred by passing fluorescent molecules through the confocal detection volume[15,23]. Photon bursts (also called spikes) in TDP-43 (FL), C220, C233, C243, and C253 were observed during photon count rate records, while not in C263, C274, and GFP (Fig. 2a). Counts per particle (CPP), which indicates fluorescence brightness per a single particle and homo-oligomeric states, of C233 was significantly higher compared to other CTFs, FL, and GFP monomers (Fig. 2b), suggesting that C233 may be efficiently homo-oligomerized. In contrast, CPPs of other TDP-43 CTFs and FL were not dramatically higher than that of GFP monomers as a control (Fig. 2b), suggesting that they may not be so effi-ciently homo-oligomerized. Next, diffusion time (DT), which corresponds to the passing speed of fluorescent molecules in the detection volume and indicates their apparent molecular weight change, of TDP-43 and its CTFs

increased compared to the GFP monomer control (Fig. 2c), indicating that TDP-43 CTFs form significantly higher molecular weight oligomers than the GFP monomers in the soluble fraction. The diffusion time of C253, C263, and C274 was slightly slower compared to other CTFs and showed a tendency to be higher than that of GFP monomers. Since the apparent molecular weight is assumed by the cube of the diffusion time ratio to the GFP monomer (27 kDa; see Eq. 2), that of C220, C233, and C243 was approximately 3.1 to 3.5 MDa, and that of C253, C263, and C274 was 0.8 to 1.7 Mda. Therefore, C220, C233, and C243 form larger soluble oligomers among FL and TDP-43 CTFs; however, C220 preferentially forms hetero-oligomers containing many endogenous molecules, while C233 tends to form relatively homogenous oligomers. Furthermore, these data did not rule out the existence of hetero-oligomers of C233 containing other cellular proteins, because the apparent molecular weight of C233 estimated from the diffusion time was much larger than that of the tetramer, while the CPP of C233 is only about four-fold higher than that of GFP monomers. Therefore, the conformation of the N-terminal region outside of GRR in TDP-43 CTFs may define the structural characteristics for the formation of oligomers with endogenous proteins.

## The 220–262 region of TDP-43 is a nucleation domain and incorporates its C-terminal glycine-rich region during its condensation

To demonstrate whether the N-terminal region defines the condensation propensity of C220, we prepared 220–262 amino acids tagged with GFP, a C-terminal portion of RRM2, of TDP-43 (F220) (Fig. 3a). When observed as independent experiments with C220 and C263 in Neuro2a cells as positive and negative controls respectively, F220 condensates were observed in the cytoplasm (Fig. 3b), while intact RRM2 condensates were not (Fig. 3b),

**Fig. 3 | Condensate formation and amyloidogenic properties of the 220–263 region of TDP-43 in Neuro2a cells. a** Domain structure and abbreviation of TDP-43 CTFs: 220–263 (F220), 191–262 (RRM2), and a mutant in which the position of F220 and C274 was swapped (SW) in addition to C220 and C263. **b** Confocal fluorescence and bright field microscopic images of Neuro2a cells expressing F220, RRM2, and SW tagged with GFP. White arrows represent cytoplasmic condensates. Bars = 5 μm. **c** The population of Neuro2a cells containing cytoplasmic condensates (conds). Bars: mean and 95% CI; Dots: raw values (*n* = 3 independent experiments). **d** Western blot of 1% SDS-soluble (S) and -insoluble (P) cell lysates using an anti-GFP and anti-α-tubulin antibody (*left*). The right values (25, 37, and 50 kDa) represent the positions of the molecular weight marker. The quantification of the abundance of TDP-43 CTFs in the insoluble (Insol.) fraction. The abundance shows the normalized band intensity in the P fraction to total (S + P) fraction (*right*). Bars: mean and 95% CI; Dots: raw values (*n* = 3 independent experiments). **e** Confocal images Neuro2a cells expressing TDP-43 CTF tagged with GFP stained with a fluorescent tracer for amyloid, Amytracker. *Middle*: A cell containing less Amytracker-stained condensates of F220. *Bottom*: A cell containing Amytracker-stained condensates of F220. Inset (*bottom*): Enlarged condensates enclosed by the white dot lines. White arrows and yellow arrowheads represent the positions of Amytracker-positive and negative condensates in the cytoplasm. The asterisk in the images represents the nucleolus. Bars = 5 μm. In **c** and **d**, *p*-values above the bars and lines: one-way ANOVA with Tukey's test compared to GFP monomers as a control and comparison between the lines, respectively. The blue color indicates no significant *p*-values. **f** Comparison of normalized relative fluorescence intensities (nRFI) of C220 and F233 tagged with GFP (magenta and green, respectively) in live Neuro2a cells using fluorescence recovery after photobleaching (FRAP). Dots and bars indicate mean ± 95% CI (*n* = 6 independent cells for C220 and *n* = 7 independent cells for F220). Inset values show the estimated mobile fraction from the maximum recovery of nRFI obtained by non-linear curve fitting analysis using a one-component exponential recovery model. The individual data points are available in Supplemental Data 1.

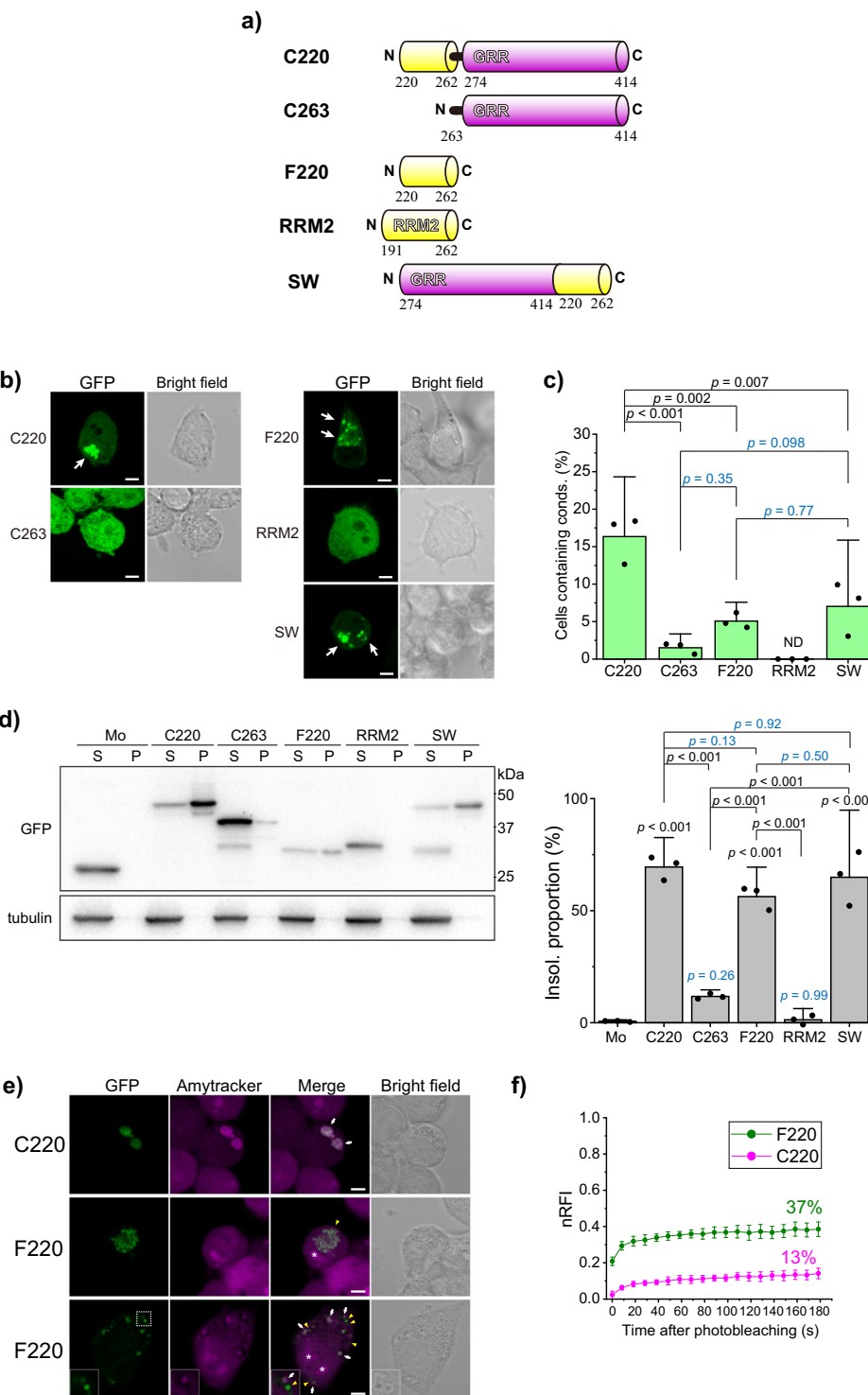

revealing that F220 itself tends to form condensates. However, the population of cells containing cytoplasmic condensates of F220 was lower than that of C220 (Fig. 3c). To demonstrate whether the position of F220 relative to GRR could be involved in the condensation propensity, we constructed a swapped mutant in which F220 was fused to the C-terminal of GRR (SW). When expressed in Neuro2a cells, we found that this construct formed cytoplasmic condensates (Fig. 3b); however, the population of cells containing these condensates was not different from that of F220 (Fig. 3c). A portion of F220 and SW was recovered in the SDS-insoluble fraction (Fraction P), while the RRM2 and GFP monomers alone were not (Fig. 3d).

For F220 and SW, more than 50% remained in the insoluble fraction as C220 (Fig. 3d, *right*) despite their condensation propensity in the cytoplasm was lower than that of C220 (Fig. 3b), suggesting that, even if they did not accumulate as condensates, they may form high molecular complexes or soluble oligomers. Consequently, F220 is an aggregation-prone peptide, and the presence of GRR on the C-terminal side of the F220 peptide promotes its condensation propensity. Next, to determine whether the deposition of TDP-43 CTFs in the cytoplasmic condensates is involved in the formation of amyloid fibrils in them, the condensates were stained with an amytracker dye, an amyloid-binding fluorescent probe. Amytracker stained the

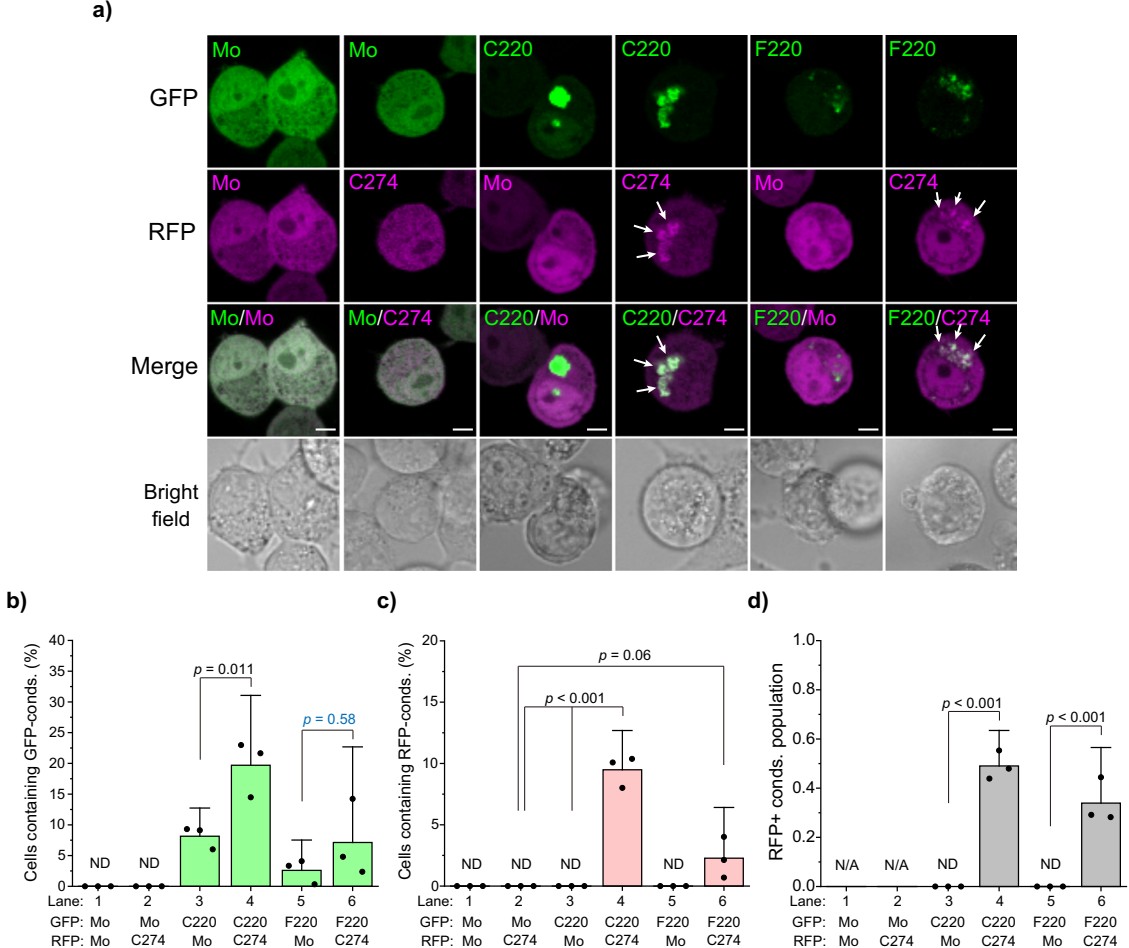

**Fig. 4 | Co-condensation of C220 and F220 with the glycine-rich region.**
**a** Confocal fluorescence and bright field microscopic images of Neuro2a cells
expressing GFP-tagged C220 or F220 with the glycine-rich region (GRR; C274)
tagged with a red fluorescent protein (RFP). Mo indicates GFP or RFP monomers.
White arrows represent C274-positive condensates in the cytoplasm. Bars = 5 μm.

**b** The population of Neuro2a cells that contain GFP-positive condensates (conds) in
the cytoplasm. **c** The population of Neuro2a cells containing RFP-positive con-
densates in the cytoplasm. **d** The ratio of RFP-positive condensates to GFP-positive
ones. In **b**, **c**, and **d**, ND: Not determined; N/A: Not applicable; $p$-values above the
lines: one-way ANOVA with Tukey's test ($n = 3$ independent experiments).

cytoplasmic condensates of C220, C233, C274, and F220 (Fig. 3e and
Supplementary Fig. 2); however, in a portion of the F220 condensates, the
fluorescence intensity of amytracker was low (Fig. 3e, *middle lines*), or the
condensates stained with amytracker and not with it coexisted (Fig. 3e,
*bottom lines*). Accordingly, F220 is prone to condensation but could have
various types of structures (i.e., amyloids and non-amyloids). However, 1,6-
hexanediol, a commonly used inhibitor of weak hydrophobic interactions,
was unable to dissolve the cytoplasmic condensates of C220, F220, and C274
(Supplementary Fig. 3), suggesting that, in the condensates, these CTFs
accumulated tightly with intermolecular entanglements, regardless of
amyloid fibril formation. Furthermore, FRAP analysis showed that the
recovery proportion of F220 in the cytoplasmic condensates was dramati-
cally higher than that of C220 (Fig. 3f). This is because the nRFI of F220 just
after photobleaching (at 0 s) was significantly higher than that of C220
(18.5% difference), but there were no significant differences in recovery
speed 20 seconds and later between F220 and C220, suggesting a low fluidity
of them in the condensates. These results suggest that, although a portion of
F220 in the condensates is strongly bound and deposited as in C220, the
other portion of F220 can move rapidly between the cytoplasm. Therefore,
we hypothesized that TDP-43 GRR as an IDR may play a key role in
determining the character of deposited as the cytoplasmic condensates.

Next, we tested whether C220 and F220 could incorporate TDP-43
GRR into the cytoplasmic condensates by co-expressing both C220 or F220
tagged with GFP (G-C220 or G-F220) and C274 tagged with a red

fluorescent protein (RFP) (R-C274) in Neuro2a cells. Confocal microscopy
of these cells revealed that condensates of RFP monomers and R-C274 were
not observed in cells expressing GFP monomers as negative controls (Fig. 4a
and b). R-C274 was colocalized with G-C220 in the cytoplasmic con-
densates, but the RFP monomers were not (Fig. 4a, c, and d). In particular,
G-F220 was colocalized with R-C274, but bot with RFP monomers (Fig. 4a,
e, and f). Cells containing condensates of GFP monomers were not detected
regardless of R-C274 expression (lanes 1 and 2 in Fig. 4b). Cells containing
G-C220 condensates were increased by co-expression of R-C274 compared
to RFP monomers as a control (lanes 3 and 4 in Fig. 4b), suggesting that
C274 promotes condensate formation of C220. Meanwhile, cells containing
G-F220 condensates did not increase much with the coexpression of
R-C274 (lanes 5 and 6 in Fig. 4b); however, R-C274-positive condensates
were observed only when G-C220 or G-F220 were co-expressed (lanes 4 and
6 in Fig. 4c). Although the cell population containing R-C274-positive
condensates in G-C220-expressing cells was the same as that in G-F220-
expressing cells (lanes 4 and 6 in Fig. 4d); therefore, the efficiency of GRR
incorporation of F220 is similar to that of C220. As a result, the 220–262
amino acid region of TDP-43 may first assemble and then condensate with
involving the GRRs, regardless of whether both the 220–262 region and
GRR are included in a single polypeptide.

Next, to determine whether TDP-43 CTFs form co-condensates with
non-artificial and physiological proteins, we tested that C220 and C233
could incorporate full-length TDP-43 into the cytoplasmic condensates by

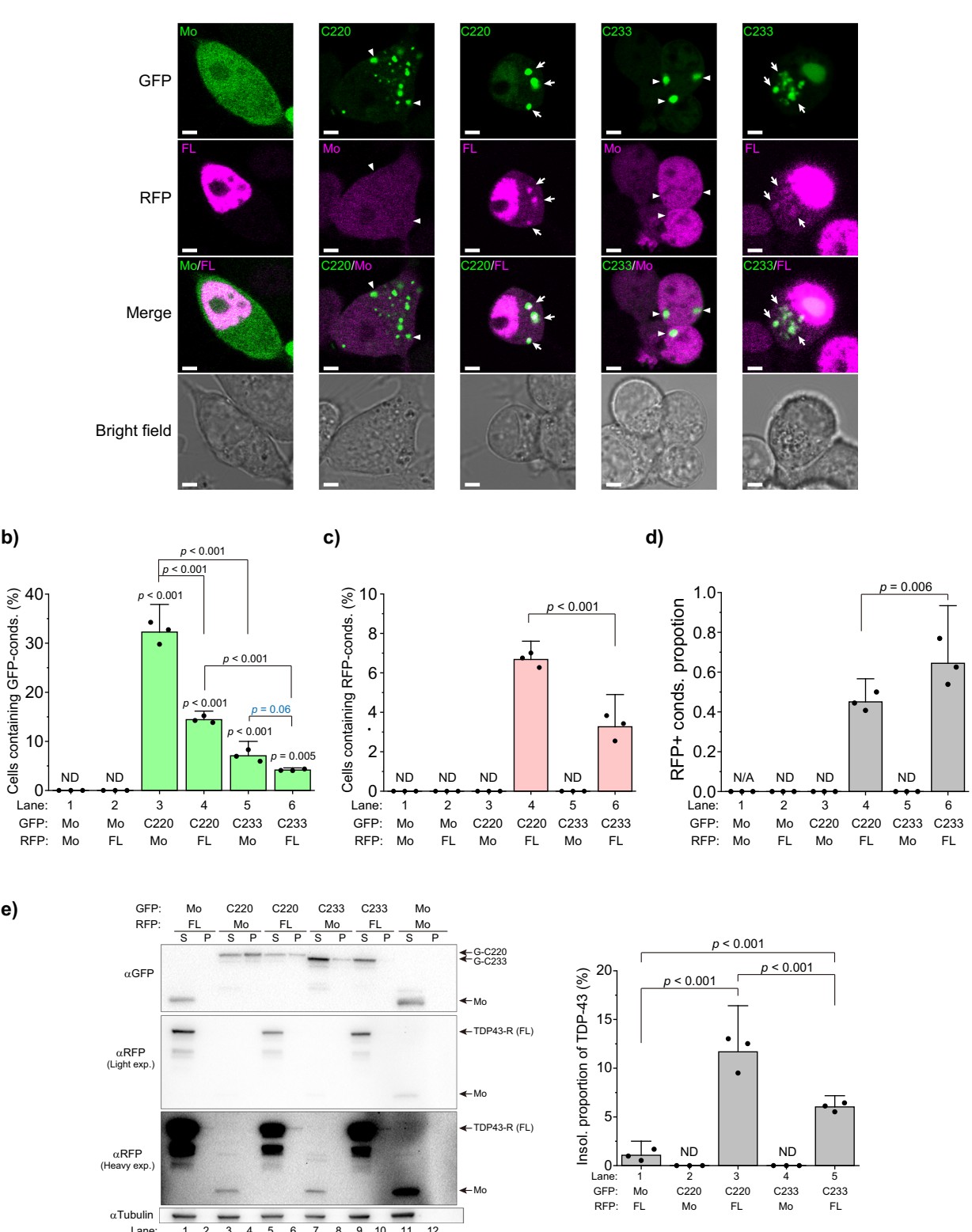

co-expressing both G-C220 or G-C233 and TDP-43 tagged with RFP (TDP43-R) in Neuro2a cells. Confocal microscopy of these cells revealed that condensates of TDP43-R were not observed in cells expressing GFP monomers as a negative control (Fig. 5a). TDP43-R was colocalized with G-C220 and G-C233 in the cytoplasmic condensates, but the RFP monomers were not (Fig. 5a). Although cells containing G-C220 and G-C233

condensates significantly decreased in TDP43-R expression compared to RFP monomer expression (Fig. 5b), cytoplasmic condensates of TDP43-R emerged (Fig. 5c) and were incorporated with G-C220 and G-233 (Fig. 5d). Furthermore, detergent-insoluble TDP43-R increased when G-C220 and G-C233 were co-expressed; however, fewer TDP43-R was observed when co-expressing G-C233 than when co-expressing C220 (Fig. 5e).

**Fig. 5 | Co-condensation of C220 and C233 with full-length TDP-43. a** Confocal fluorescence and bright field microscopic images of Neuro2a cells expressing GFP-tagged C220 or C233 with TDP-43 (FL) tagged with a red fluorescent protein (RFP) (TDP43-R). Mo indicates GFP or RFP monomers. White arrowheads represent TDP-43-positive condensates in the cytoplasm. White arrows represent TDP-43-negative condensates in the cytoplasm. Bars = 5 μm. **b** The population of Neuro2a cells that contain GFP-positive condensates (conds) in the cytoplasm. **c** The population of Neuro2a cells containing RFP-positive condensates in the cytoplasm. **d** The ratio of RFP-positive condensates to GFP-positive ones. ND: Not determined;

N/A: Not applicable; *p*-values above the lines: one-way ANOVA with Tukey's test (*n* = 3 independent experiments). **e** Western blot of 1% SDS-soluble (S) and -insoluble (P) cell lysates using an anti-GFP, anti-RFP, and anti-α-tubulin antibody (*left*). The right arrows represent the migration position of the proteins. GFP monomers: Mo. The quantification of the abundance of TDP43-R in the insoluble (Insol.) fraction. The abundance shows the normalized band intensity in the P fraction to the total (S + P) fraction (*right*). Bars: mean and 95% CI; Dots: raw values; ND: Not determined; *p*-values above the lines: one-way ANOVA with Tukey's test (*n* = 3 independent experiments).

Consequently, C220 and C233 condensates are able to incorporate full-length native TDP-43 depending on the degree of the condensation. Moreover, C220 and C233 condensate and/or their molecular species before condensation may behave like a prion.

### The hydrophobic region in the intrinsically disordered region of TDP-43 CTF drives protein condensation

Why is the proportion of cells containing C220 and F220 condensates dramatically increased by C274 co-expression, as shown in Fig. 4b? The hydrophobic interaction is known to be important for the stabilization of the liquid condensates of TDP-43[54], and the hydropathy plot confirmed that hTDP-43 311–340 is a highly hydrophobic region (HR; Supplementary Fig. 4). Nuclear magnetic resonance (NMR) analysis of hTDP-43 311–360 peptide in solution determined that α-helical structure (amino acids 321–340) is included in this HR despite being in the IDR[55]; however, 282–360 amino acid region including the helical structure is converted to cross-β amyloid filaments in the pathological structure from ALS with FTLD[38]. The α-helical structure in GRR is proposed to be involved in homotypic interactions of TDP-43 CTFs in the condensates with tuning liquid-liquid phase transition[56,57]. Therefore, we determine whether the efficiency of condensate formation is characterized by the HR in GRR using a variant of G-C220 that lacks HR (C220Δ). Furthermore, because TDP-1, a TDP-43 orthologue of *C. elegans*, has a short and non-conserved C-terminal region compared to human TDP-43 GRR[55], we used a chimeric protein that F220 was fused to the C-terminal region of TDP-1 (F220c) (Fig. 6a). Although C220Δ and F220c formed the cytoplasmic condensates in Neuro2a cells (Fig. 6b), the proportion of cells containing condensates of C220Δ and F220c was significantly lower than that of C220, and it was not different from that of F220 (Fig. 6c). Next, to determine the solubility of such CTFs, the amount of the CTFs in SDS-soluble (S) and -insoluble (P) fractions of cell lysates was analyzed using SDS-PAGE followed by western blotting (Fig. 6d). A portion of C220Δ was recovered in the P fraction as similar to F220 and C220, suggesting that the condensates of C220Δ in the cytoplasm may tightly assemble as aggregates but the assembly state is similar to F220. However, almost all F220c was recovered in the S fraction. The population of dead cells expressing C220Δ and F220c was lower than that of C220, and it was not different from that of F220 (Fig. 6d). There was a correlative relationship between cytoplasmic condensate formation and dead cell proportion. Condensation of a GFP-tagged TDP-1 CTF (295–414 amino acids; TDP1-C295), corresponding to the human C220 sequence, was also analyzed. Unlike F220c, TDP1-C295 showed properties similar to C220 in Neuro2a cells (Supplementary Fig. 5a–e). These results suggest that the combination of both a portion of RRM2 and the C-terminal disordered region that are evolutionarily conserved may play an important role in the condensation of TDP-43 CTFs, and that HR including α-helical structure in GRR may be involved in the structural transition into tightly assembled aggregates.

To determine whether HR in the GRR is required for the molecular assembly of TDP-43 CTFs in the condensates, HR-deleted C274 (C274Δ) was prepared and analyzed condensation properties (Fig. 7a). Observation of the cytoplasmic condensation in Neuro2a cells using confocal fluorescence microscopy showed that, although colocalization of G-C220 and R-C274 in the cytoplasmic condensates was confirmed as shown in Fig. 4, that of G-C220 and RFP-tagged C274Δ (R-C274Δ) was not (Fig. 7b). On the

contrary, R-C274 was co-localized with GFP-tagged C220Δ (G-C220Δ) in the condensates, while RFP monomers as a negative control were not (Fig. 7b). The proportion of cells containing G-C220 condensates was slightly decreased when the co-expressing partners were changed from R-C274 to R-C274Δ (lanes 4 and 5 in Fig. 7c), but still higher than that when RFP monomers were expressed (lanes 2 and 5, Fig. 7c). However, R-C274Δ was not incorporated into the G-C220 condensates (lane 5 in Fig. 7d and e). This is because, since C274Δ does not contain HR but contains F220, this region may promote the C220 condensation through the intact GRR of the counter polypeptide. The SDS solubility assay showed that, although R-C274 was observed in the P fraction of the cell lysates when G-C220 was co-expressed, R-C274Δ in the P fraction was dramatically decreased and dim (lanes 6 and 8 in Fig. 7f; lanes 3 and 4 in Fig. 7h).

Next, we investigate whether C274 including HR could be incorporated into the condensates of C220 lacking HR (C220Δ). The cells containing G-C220Δ condensates were increased by co-expression of R-C274 compared to the expression of RFP monomers as a negative control (lanes 3 and 6 in Fig. 7c), and this increase was more notable than the increase of the cells containing condensates of G-F220 by co-expression of R-C274 (lanes 5 and 6 in Fig. 4b). Co-localization between G-C220Δ and R-C274 in the condensates was observed despite the low efficiency of condensate formation and their co-localization compared to co-expression with HR-including C220 (G-C220) (lanes 4 and 6 in Fig. 7d, e). The SDS solubility of these TDP-43 CTFs in the cell lysates showed that R-C274 was observed in the P fraction when G-C220 or G-C220Δ was co-expressed, but R-C274Δ was not (lanes 6, 8, and 12 in Fig. 7f; lanes 3, 4, and 6 in Fig. 7h). Moreover, G-C220 and G-C220Δ observed in the P fraction did not change when co-expressing R-C274 or RFP monomers (Fig. 7g), suggesting that detergent-insoluble TDP-43 CTFs may not completely form cytoplasmic condensates that have enough size to be observed in fluorescence microscope but exist as high molecular weight species that are enough to be precipitated. Therefore, HR in GRR is required not only for the intra but also for intermolecular assembly of TDP-43 CTFs in the condensates after the aggregation of its N-terminal side.

Next, we investigated whether the HR region in TDP-43 GRR is involved in the condensation of full-length TDP-43. However, since wild-type TDP-43 does not form cytoplasmic condensates in Neuro2a cells[23], we used an aggregate-prone mutant of TDP-43 (C173/175S and mutation in nuclear localization signal sequence; TDP43CS) (Supplementary Fig. 6a) and expressed in human embryonic kidney 293 cells. Cells containing condensates of TDP43CS lacking HR (TDP43CSΔ) slightly decreased compared to that of TDP43CS (from ~3 to ~2%; Supplementary Fig. 6b). We tried to compare the abundance of high molecular weight species of TDP43CS and TDP43CSΔ using filter retardation assay[15,58]; however, the amount of TDP43CSΔ retarded on the membrane was not different from that of TDP43CS (Supplementary Fig. 6c). This is likely because the condensation propensity in the cell is low, and therefore we conclude that it is not able to detect small changes in the amount of high molecular weight species of TDP43CS and TDP43CSΔ. Thus, to investigate the role of HR in full-length TDP-43, we observed co-condensation between G-C220 and RFP-tagged TDP-43 lacking HR (TDP43Δ-R; Supplementary Fig. 7a) in Neuro2a cells using fluorescence confocal microscopy as in Fig. 5. TDP43Δ-R was not observed in the cytoplasmic condensates of G-C220 typically (Supplementary Fig. 7b). Although cells containing G-C220 condensates

**Fig. 6 | The condensation and cytotoxicity of TDP-43 CTFs that carry hydrophobic region-lacking GRR in Neuro2a cells. a** Domain structure and abbreviation of *Homo sapiens* TDP-43, *Caenorhabditis elegans* orthologue of TDP-43 (TDP-1), and its CTFs: amino acid 220-414 (C220), 220-262 (F220), C220 lacking 311-340 region (C220Δ), and a fusion protein of F220 and C-terminal region (C-term) of TDP-1 (F220c). **b** Confocal fluorescence and bright field microscopic images of Neuro2a cells expressing GFP-tagged C220Δ and F220c. Cells containing cytoplasmic condensates (*bottom*) or not containing them (*top*) were represented. Bars = 5 μm. **c** The population of Neuro2a cells containing condensates (conds) in the cytoplasm. Mo denotes GFP monomers. Bars: mean and 95% CI; Dots: raw values (*n* = 4 independent experiments). **d** *Left*: Western blot of 1% SDS-soluble (S) and -insoluble (P) cell lysates using an anti-GFP and anti-α-tubulin antibody. The right values (25, 37, and 50 kDa) represent the positions of the molecular weight marker. *Right*: The quantification of the abundance of TDP-43 CTFs in the insoluble fraction (Insol.). The abundance shows the normalized band intensity in the P fraction to total (S + P) fraction (*n* = 3 independent experiments) (right). **e** The proportion of dead cells expressing GFP-tagged C220, F220, C220Δ, and F220c. In *C* and *D*, Bars: mean and 95% CI; Dots: raw values (*n* = 4 independent experiments); ND: Not determined; *p*-values above the lines: one-way ANOVA with Tukey's test.

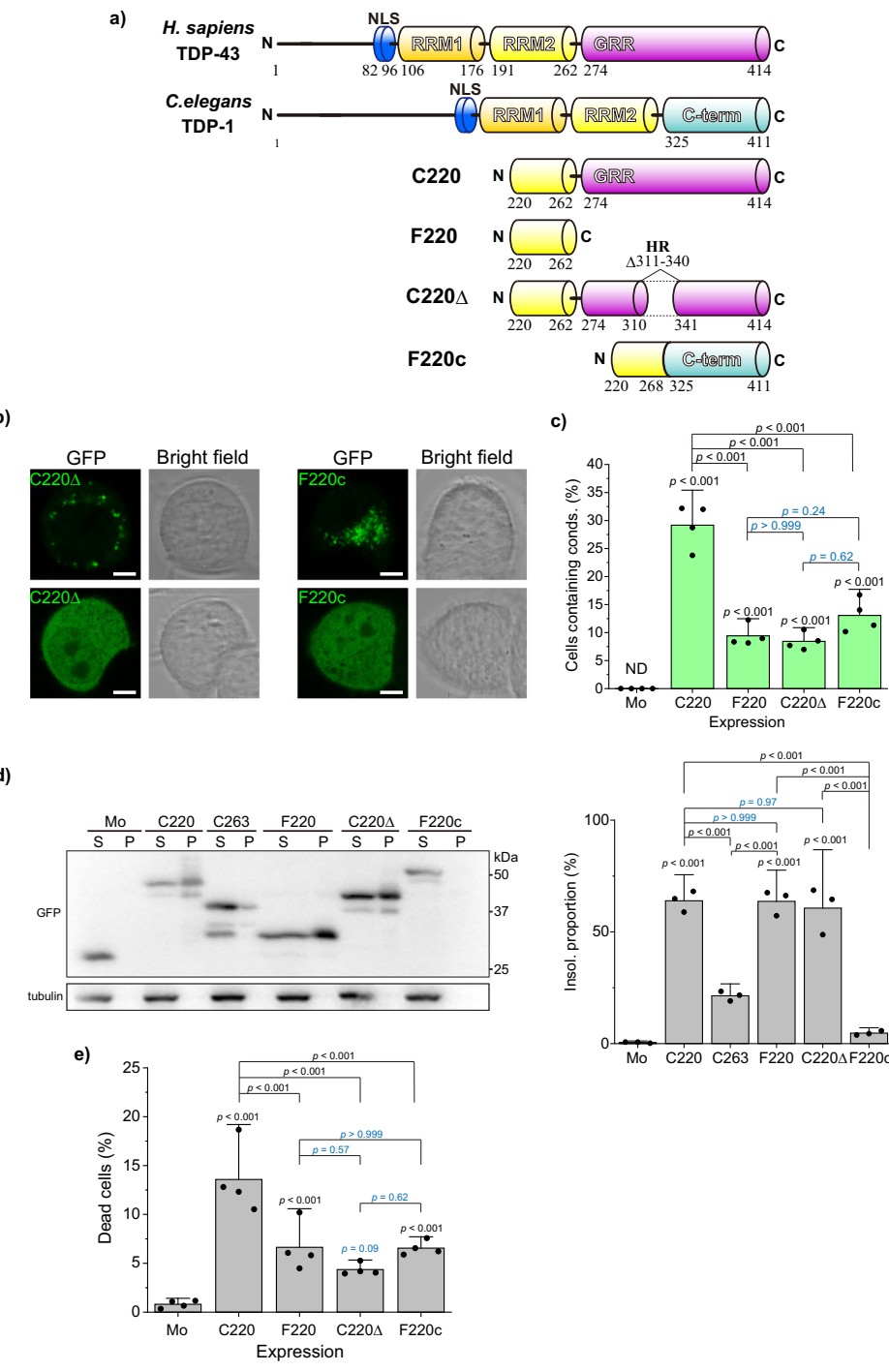

slightly increased when co-expressing TDP43Δ-R compared to when co-expressing TDP43-R, both cells containing TDP43Δ-R condensates and its proportion that was colocalized in G-C220 condensates were lower than that of TDP43-R as control (Supplementary Fig. 7c). These results suggest that HR in TDP43 and its CTFs is required for condensation and molecular assembly; moreover, C220 plays a role in seeding of aggregation and its prion-like behavior possibly transmits to intact TDP-43 through HR region.

### N- or C-terminal proximity analysis of TDP-43 CTFs in live Neuro2a cells using a blue light-induced oligomerization system

To investigate the promotion of oligomerization of TDP-43 CTFs when the N-terminal side becomes proximal, a blue-light-induced homo-oligomerization tag engineered from a plant cryptochrome variant,

CRY2olig (CRY), was used[59]. CRY was tagged to C220, C220Δ, C263, and C263Δ at their N-terminal end. To visualize the oligomerization in live cells, RFP was also tagged at the C-terminal end (CRY-C220-R, CRY-C220Δ-R, CRY-C263-R, and CRY-C263Δ-R, respectively), and expressed in Neuro2a cells. As a control, all cells expressing RFP-tagged CRY (CRY-R) formed foci in the cytoplasm and nucleoplasm by blue-light irradiation (Fig. 8a, a; lane 1 in Fig. 8b). CRY-C220-R foci were not observed in all cells after the light irradiation; however, a low population of cells expressing HR-deleted C220 (CRY-C220Δ-R) formed foci after the light irradiation (Fig. 8a–c; lanes 2 and 3 in Fig. 8b). On the contrary, almost all cells (>90%) expressing CRY-C263-R or CRY-C263Δ-R significantly formed foci after the light irradiation (Fig. 8a, d, e; lanes 4 and 5 in Fig. 8b). These results suggest that the N-terminal side of TDP-43 GRR may be easy to close regardless of HR

(Fig. 8c); however, C220 may have a steric hindrance with difficulties in intermolecular proximity at its N-terminus.

Next, to investigate the promotion of oligomerization of TDP-43 CTFs when the C-terminal side becomes proximal, CRY-R was tagged to C220,

C220Δ, C263, and C263Δ at their C-terminal end (C220-CRY-R, C220Δ-CRY-R, C263-CRY-R, and C263Δ-CRY-R, respectively). As a negative control, a variant tagged with GFP on the N-terminal side of CRY-R (G-CRY-R) was used. G-CRY-R formed foci in all cells similar to CRY-R after

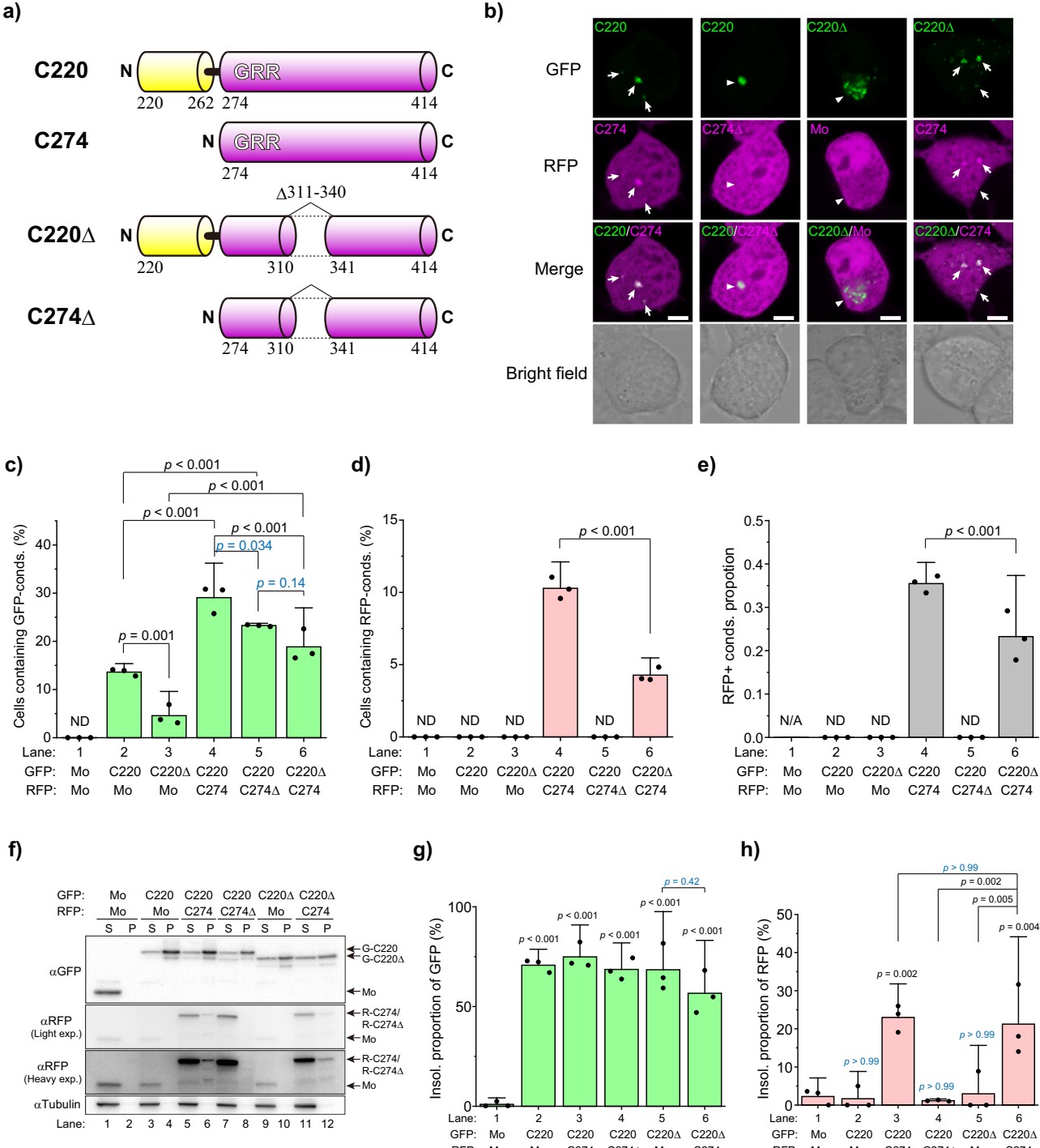

**Fig. 7 | The role of the hydrophobic region in the glycine-rich region of TDP-43 to the co-condensation process in Neuro2a cells. a** Domain structure and abbreviation of TDP-43 CTFs: amino acids 220–414 (C220), 274–414 (C274), C220 lacking 311–340 region (C220Δ), and C274 lacking 311-340 region (C274Δ). GRR denotes a glycine-rich region. **b** Confocal fluorescence and bright field microscopic images of Neuro2a cells co-expressing GFP- and RFP-tagged TDP-43 CTFs. Mo denotes GFP or RFP monomers. White arrows represent cytoplasmic condensates including both GFP- and RFP-tagged CTFs. White arrowheads represent cytoplasmic condensates including only GFP-tagged CTFs. Bars = 5 μm. **c, d** The population of Neuro2a cells

containing GFP-positive and RFP-positive cytoplasmic condensates (conds). **e** Ratio of cells containing RFP-positive condensates in a group of cells containing GFP-positive condensates. **f** Western blot of 1% SDS-soluble and -insoluble cell lysates using anti-GFP or anti-RFP antibodies. **g** The quantification of the abundance of GFP-tagged proteins in the insoluble fraction (Insol.). **h** The quantification of the abundance of RFP-tagged proteins in the insoluble fraction (Insol.). **c, d, e, g,** and **h** Bars: mean and 95% CI; Dots: raw values (*n* = 3 independent experiments); ND: Not determined; N/A: Not applicable; *p*-values above the lines: one-way ANOVA with Tukey's test.

**Fig. 8 | Intermolecular assembly of C220 and C263 using a blue light-induced oligomerization tag derived from a cryptochrome variant in Neuro2a cells. a** Fluorescence images of Neuro2a cells expressing TDP-43 CTFs: C220, C220Δ, C263, and C263Δ tagged with a cryptochrome-derived oligomerization inducer, CRY2olig (CRY), and RFP before and after the 488 nm-light irradiation. The order of the hyphens before and after indicates the N/C-terminal side of the CRY tag. White arrowheads represent cells with foci formation after the irradiation. Bars = 5 μm. **b** Population of cells that form foci after the blue-light irradiation obtained from all measured cells (the numbers of independently counted cells were 26 for Lane 1, 27 for Lane 6, 36 for Lanes 2, 5, and 7, 35 for Lanes 3, 4, 10, and 34 for Lanes 8 and 9). ND: Not determined; *p*-values above the bars and lines: hypothesis test for the difference in the population proportions compared to a tag itself as control and comparison between the lines, respectively; ****p* < 0.001. **c** Schematic diagram of CRY tagged at the N-terminus of TDP-43 CTFs approaching and assembling after light irradiation. **d** Schematic diagram of CRY tagged at the C-terminus of TDP-43 CTFs approaching but not assembling even after light irradiation.

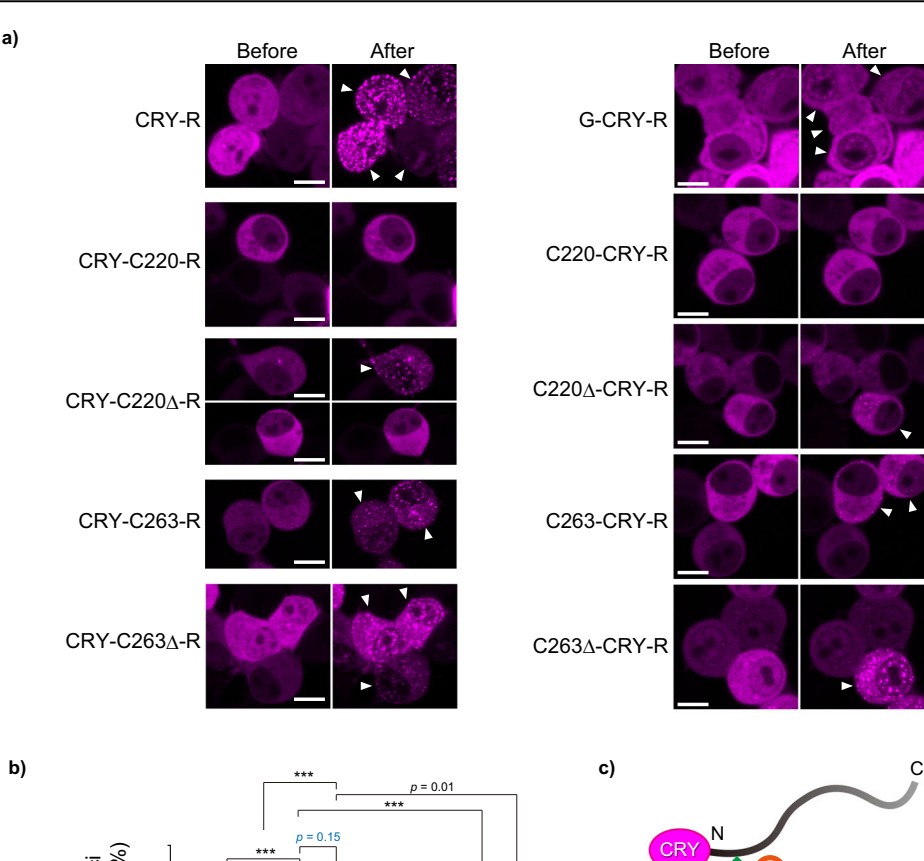

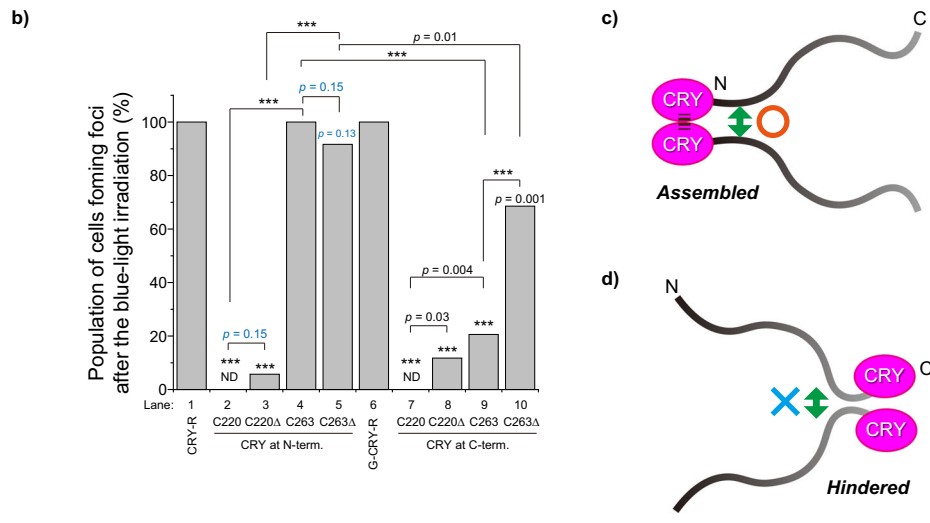

the light irradiation (Fig. 8a, f; lane 6 in Fig. 8b). Similarly to the N-terminal induction of oligomerization in C220, C220-CRY-R foci were not observed entirely after the light irradiation; however, a low population of cells expressing HR-deleted C220 (C220Δ-CRY-R) formed foci after the light irradiation (Fig. 8a, g, h; lanes 7 and 8 in Fig. 8b). Unlike CRY-C263-R or CRY-C263Δ-R, the foci-forming population of the cells expressing C263-CRY-R and C263Δ-CRY-R was dramatically low; however, C263Δ-CRY-R showed significantly increased foci-forming cells compared to C263-CRY-R (Fig. 8a, i, j; lanes 9 and 10 in Fig. 8b). These results suggest that the C-terminal side of TDP-43 GRR, unlike its N-terminal side, may have a steric hindrance with difficulty in intermolecular proximity (Fig. 8d). Furthermore, HR deletion could weaken the steric hindrance of the C-terminal side of GRR, suggesting that HR and α-helical structure in it may help to maintain a relatively rigid conformation even in the IDR.

We aimed to investigate biological consequences after the light-induced condensation of CRY-C263-R in the cells. While G-C220 and G-C274 condensates in the cytoplasm contained amyloid (Fig. 3e and Supplementary Fig. 2), the condensates CRY-C263-R, CRY-C263Δ-R, C263-CRY-R, and C263Δ-CRY-R did not (Supplementary Fig. 8a). Cells

expressing CRY-C263-R, CRY-C263Δ-R, C263-CRY-R, and C263Δ-CRY-R at 1 h after allowing blue light irradiation remained viable (Supplementary Fig. 8b). Consequently, we did not observe toxic and amyloidogenic conversion of CRY-induced TDP-43 GRR condensates. To confirm the persistence of the CRY-induced condensates of TDP-43 GRR, we assessed the population of cells containing cytoplasmic condensates 20 minutes after light irradiation. A significant proportion of cells preserves the condensates of CRY-C263-R, CRY-C263Δ-R, and CRY-R, with the population containing CRY-C263-R being particularly high (~80%). However, the population of the cells containing that of C263-CRY-R, C263Δ-CRY-R, and G-CRY-R was either negligible or very low (Supplementary Fig. 8c). The half disassembly time of CRY-C263-R and CRY-C263Δ-R condensates was slower than that of CRY-R, with CRY-C263-R exhibiting the slowest disassembly kinetics (Supplementary Fig. 8d). These results indicate that the CRY-induced TDP-43 GRR condensates can be disassembled, but TDP-43 IDR may play a role in retarding the condensate disassembly kinetics. At 8 h after light irradiation, only a small number of cells contained foci of CRY-R and CRY-C263-R, but these foci were not stained with amytracker (Supplementary Fig. 8e). Consequently, CRY-C263 can investigate steric

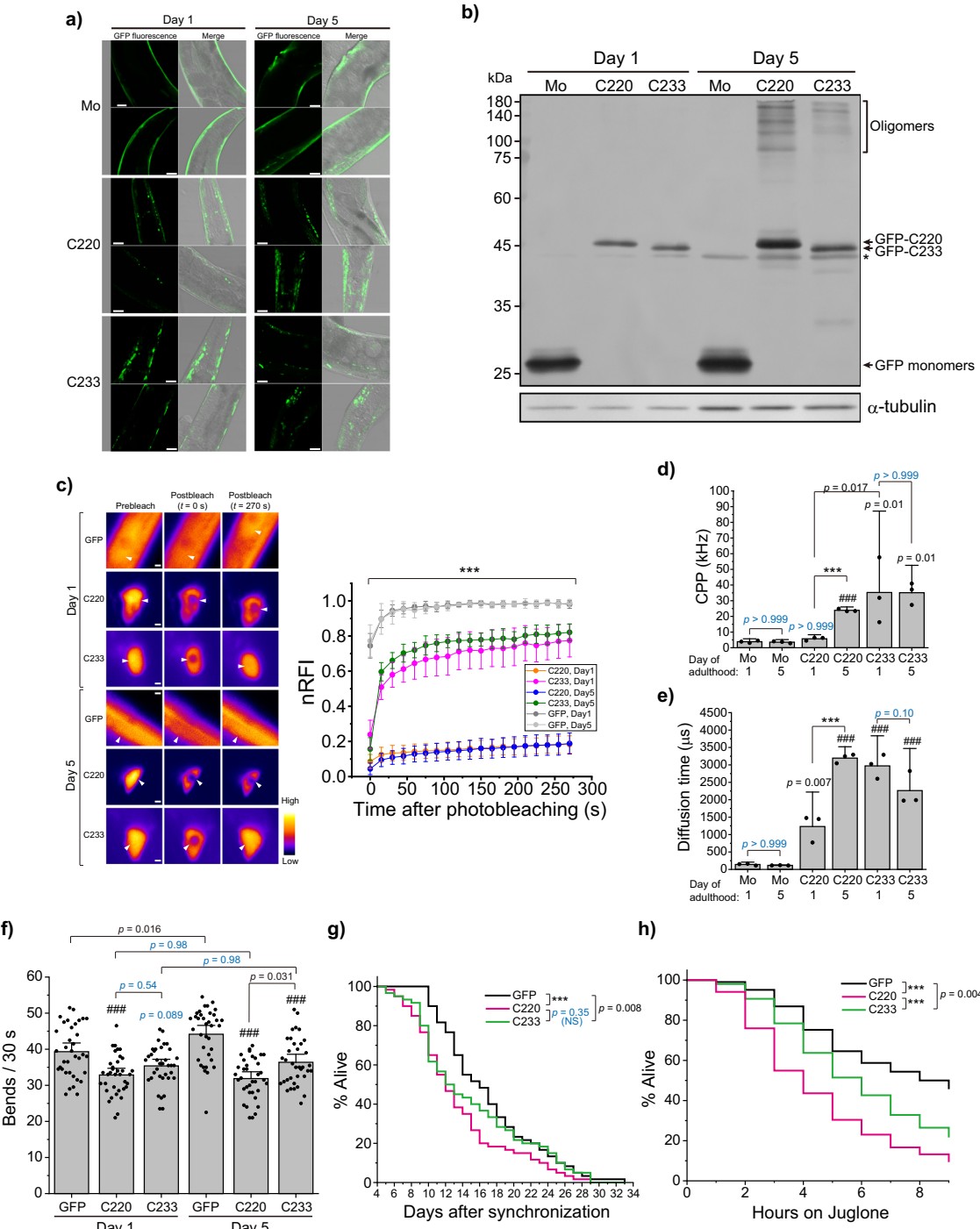

hindrance when molecular chains are in close proximity, but does not lead to toxic condensation, at least during short periods of light irradiation in a immortalized cell line.

## Condensation properties of GFP-tagged C220 and C233 in nematodes

To demonstrate proteotoxicity of TDP-43 CTFs, we established a new *C. elegans* model expressing G-C220 or G-C233 in body-wall muscle. Unlike Neuro2a cells, G-C233 condensates were observed in all fluorescence-positive worms as same as G-C220 on both 4 and 8 days after synchronization by bleaching (hereafter, days 1 and 5 of adulthood, respectively), while GFP monomers did not form any condensates (Fig. 9a and Supplementary Fig. 9). The age-dependent change in expressing amounts of GFP

monomers, G-C220, and G-C233 was analyzed using western blotting (Fig. 9b). The expression levels of GFP monomers did not change from day 1 to 5 of adulthood. However, the amount of G-C220 increased, whereas that of G-C233 did not, from day 1 to 5 of adulthood. Furthermore, high molecular weight species of G-C220 and G-C233 emerged on day 5, and it was more prominent in G-C220 than G-C233, suggesting that tightly assembled oligomers of G-C220 and G-C233 may be accumulated in the worms as ageing.

The fluidity of G-C220 and G-C233 in the condensates in live worms was compared using FRAP. On both days 1 and 5 of adulthood, the maximum recovery of nRFI of GFP monomers as a freely mobile control was ~100% (Fig. 9c). The maximum recovery of nRFI of G-C220 was less than 20%, while that of C233 was more than 75% fluorescence recovery within

**Fig. 9 | Characteristics of TDP-43 C-terminal fragments in nematodes. a** Confocal fluorescence and bright field images of nematodes expressing GFP-C220, GFP-C233, and GFP monomers (Mo) on 4 or 8 days after synchronization by bleaching (Day 1 and 5 of adulthood, respectively). The two images of the same sample are from the same nematode. The low magnification images of the nematodes displayed here are shown in Supplementary Fig. 9. Bars = 20 μm. **b** Western blot of nematode lysates in 4 or 8 days after synchronization by bleaching (Day 1 and 5, respectively) using an anti-GFP and anti-α-tubulin antibody. Arrows show the monomer-corresponding bands for GFP-C220, GFP-C233, and GFP monomers. Oligomers indicate high molecular weight species. The asterisk shows a non-specific staining band by anti-GFP antibody. **c** Fluorescence recovery after photobleaching (FRAP) analysis of GFP monomers as a control and condensates of C220 or C233 tagged with GFP in live nematodes on 4 or 8 days after synchronization by bleaching (Days 1 and 5 of adulthood, respectively) *Left*: Typical fluorescence images of condensates in nematodes before and after the photobleaching. Arrowheads indicate photo-bleached positions. Bars = 0.5 μm. Pseudocolor indicates the fluorescence intensities. *Right*: Plots of the fluorescence recovery curve after the photobleaching of GFP monomers, GFP-C220, and -C233 in live nematodes (mean ± 95% CI; $n = 7$ independent experiments). The Y-axis represents the mobile fraction (normalized relative fluorescence intensity; nRFI) of C220 and C233 in the condensates and GFP monomers. Student's *t*-test for comparison among GFP monomers, C220, and C233 at the same day: ***$p < 0.001$. Statistical comparison just after the photobleaching ($t = 0$ s) is represented in Supplementary Fig. 10. The individual data points are

available in Supplemental Data 1. **d, e** Fluorescence correlation spectroscopy (FCS) analysis of GFP-C220, -233, and GFP monomers (Mo) in nematode lysates on 4 or 8 days after synchronization by bleaching (Day 1 and 5). Counts per particle (CPP) in **d** and diffusion time in **e**. Bars indicate mean and 95% CI ($n = 3$ independent experiments). *p*-values above the bars and lines: one-way ANOVA with Tukey's test compared to GFP monomers as control and comparison between the lines (###$p < 0.001$ and ***$p < 0.001$), respectively. NS with a cyan color indicates $p > 0.05$. **f** The number of bends per 30 seconds of nematodes expressing GFP-C220, -C233, and GFP monomers on 4 or 8 days after synchronization by bleaching (Day 1 and 5 of adulthood, respectively) in M9 buffer. Bars indicate the mean and 95% CI of 36 nematodes. *p*-values above the bars and lines: one-way ANOVA with Tukey's test compared to GFP monomers as control and comparison between the lines (###$p < 0.001$ and ***$p < 0.001$), respectively. NS with cyan color indicates $p > 0.05$. **g** Survival curve of synchronized nematode expressing GFP-C220, -C233, or GFP monomers (magenta, green, and dark gray, respectively) ($n = 75$ nematodes). The *x*-axis shows the day after the synchronization by bleaching (i.e., its day 4 corresponds to day 1 of adulthood). *p*-values were obtained from Gehan's generalized Wilcoxon test: ***$p < 0.001$. NS indicates $p > 0.05$. **h** Survival curve of synchronized nematode expressing GFP-C220, -C233, or GFP monomers (magenta, green, and dark gray, respectively) when 270 μM Juglone were treated (n = 200 nematodes). The *x*-axis shows time on Juglone-containing plates. *p*-values were obtained from Gehan's generalized Wilcoxon test: ***$p < 0.001$.

270 s (Fig. 9c), suggesting that the fluidity of C233 in the condensates was high but not C220. The RFI just after photobleaching showed that a portion of G-C233 was highly liquid-like and rapidly moving in the condensates compared to G-C220 on days 1 and 5 of adulthood (Supplementary Fig. 10). Next, to determine the oligomeric states of G-C220 and G-C233 in the worms, CPP and diffusion time of them using FCS were analyzed in the soluble fraction of worm lysates (Fig. 9d, e). The CPP of G-C220 on day 1 was similar to that of GFP monomers but dramatically increased on day 5, suggesting that G-C220 may form homo-oligomers on day 5 of adulthood. On the contrary, the CPP of G-C233 was significantly high on both days 1 and 5 compared to GFP monomers and G-C220 on day 1, and its large variance on day 1 became smaller on day 5, suggesting that G-C233 may form homo-oligomers on both days 1 and 5, but from day 1 to 5, the stoichiometry of C233 may converge to more uniform one. Diffusion times of G-C220 and G-C233 were significantly increased compared to GFP monomers on both days 1 and 5. Although the diffusion time of G-C233 was not different from day 1 to 5, that of G-C220 became dramatically longer. Consequently, G-C220 had a higher percentage of hetero-oligomers with endogenous other proteins than homo-oligomers on day 1 of adulthood; however, G-C220 on day 5 and G-C233 on days 1 and 5 of adulthood contained homo-oligomers significantly, although hetero-oligomers with endogenous proteins could be co-existed because diffusion times of G-C220 and -C233 are dramatically higher than the CPP ratio for GFP monomers as observed in the lysate of Neuro2a cells.

**Comparison of proteotoxicity between C220 and C233 in the nematodes**

To investigate the proteotoxic effects of G-C220 and G-C233 in the worms, locomotion behavior and lifespan of worms were analyzed (Fig. 9f and g, respectively). The bending activity of the worms expressing G-C220 was low on day 1 compared to those expressing GFP monomers as a control, while that of G-C233 was slightly low but not prominent (Fig. 9f). From day 1 to 5, the bending activity of worms expressing GFP monomers was increased, indicating the progression of locomotion behavior by their body growth as aging; however, such a progression was not observed in worms expressing G-C220 and G-C233. Moreover, on day 5, there is a contrast in the bending activities between G-C220 and G-C233 because that of G-C220 decreased compared to day 1 but G-C233 did not. Accordingly, both G-C220 and G-C233 affect locomotion behavior; however, G-C220 affects it more significantly.

The lifespan of worms expressing G-C220 and G-C233 (Median life expectancy: 8.4 and 9.0 days after synchronization by bleaching,

respectively) was then significantly shortened compared to worms expressing GFP monomers as a control (Median life expectancy: 12.5 days after the synchronization) (Fig. 9g). Although the mortality of the worms expressing G-C220 and G-C233 were similar during young to middle adults (from day 4 to 13 after the synchronization), that of G-C220 decreased during old adults (after day 13 after the synchronization), but this difference between C220 and C233 was not significant (Fig. 9g). Given that TDP-43 aggregation is triggered by oxidative stress in ALS[60], and that the lifespan of worms exposed to Juglone, a producer of intracellular oxide levels[61], can be shortened, we would expect that worms expressing TDP-43 CTFs would display increased sensitivity to Juglone-induced toxicity. This aligns with the toxicity profile of TDP-43 CTFs, such as C220 versus C233. The lifespan of worms treated with Juglone was analyzed in a dose-dependent manner. When 270 μM Juglone was treated, all worm strains were gradually dead, but the lifespan of G-C233-expressing worms was significantly longer than G-C220-expressing one but shorter than GFP-expressing one (Fig. 9h). These results suggest that C220 has a stronger effect on shortening lifespan than C233 in an oxidative stress condition; i.e., C220 may be more toxic than C233.

Accordingly, C220 and C233 expression showed significant proteotoxic effects resulting in a cell-nonautonomous manner in the worms, but that of C233 was mild compared to C220.

## Discussion

TDP-43 CTFs are known to form condensates in the cell[22,26]. Here, we showed that the 220–262 region of tRRM2 on the N-terminal side of the GRR has an important role in the formation of the condensates. The intact RRMs of TDP-43 can be folded stably because the solution and crystal structures of recombinant RRM1, RRM2, and a tandem of them expressed in *E.coli* are determined[62–64]. A 35 kDa TDP-43 CTF containing two intact RRMs (RRM1 and RRM2) and GRR, TDP35 (amino acids 90–414), showed a low propensity for the cytoplasmic condensation and was not toxic because the cell death propensity of TDP35 was as same as that of TDP-43[23], in addition to, unlike TDP25/C220, no colocalization between the TDP35 condensates and ubiquitin, suggesting that it would not include misfolded one[23]. Consequently, a stably folded domain at the N-terminal side of the GRR could stabilize the conformation of the CTFs and then be less likely to form condensates containing misfolded ones. On the other hand, peptides in a portion of tRRM2 (amino acids 208–265 and 234–273) are reported to be less stable and form amyloid fibrils in vitro[41,65]. The cytoplasmic condensates of F220, the tRRM2 region of TDP25/C220, did not necessarily consist only of amyloid fibrils (Fig. 3). Although RRM2 contains two α-

**Fig. 10 | Model of structural change and aggregate formation of TDP-43 carboxy-terminal fragments. a** A scheme of intermolecular assembly and aggregate propagation of TDP-43 CTFs that includes truncated RRM2 (tRRM2) and GRR containing hydrophobic region (HR). **b** Domain structure of RRM2 and glycine-rich region (GRR) of TDP-43 and a putative model for conformational change of C220. The secondary structure of human RRM2 of TDP-43 is homologically predicted from the solution NMR structure of murine RRM2 (PDB#3D2W): α1 and α2 indicate the first and second α-helix structure in RRM2; β1–5 indicate the five β-strands, respectively. h indicates α-helical structure (322–342) from solution NMR structure of a C-terminal fragment of human TDP-43 (PDB#2N3X). Green β-strands represent potentially amyloidogenic sequences.

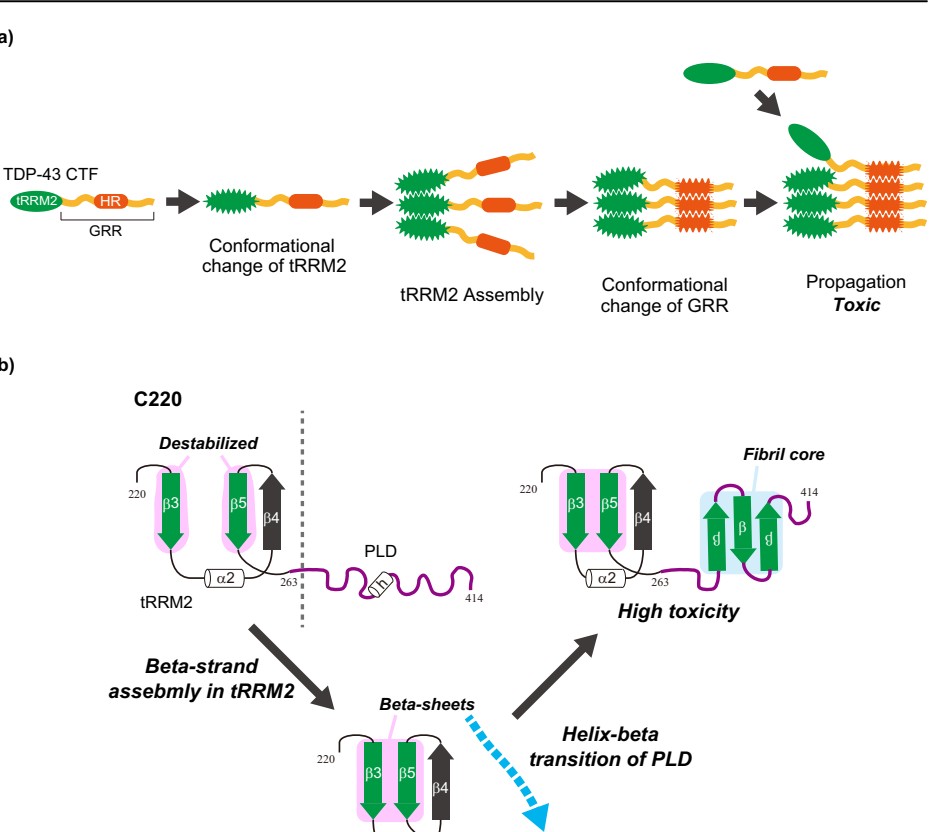

helixes (α1–2) and five β-strands (β1–5) (Supplementary Fig. 11a), TDP25/C220 lacks α1, β1, and β2. After the lack of α1, β1, and β2 by the cleavage of RRM2, it is proposed that the remaining β-strands (β3, β4, and β5) in the RRM2 are abnormally exposed because β1 was sandwiched between β3 and β5 in the intact RRM2[41]; then, β3 and β5 may be destabilized (Fig. 10b and Supplementary Fig. 11b). As a result, it is thought that exposed β3 and β5 are thought to be lead to misfolding of tRRM2 and may provide an opportunity to form intermolecular assembly (e.g., oligomerization) (Fig. 10b).

Amyloid fibril structures of the β-strand-transferred GRR of TDP-43 were identified in patients with ALS and FLTD patients and as recombinant ones using cryo-EM[37,38]. How do such misfolded protein oligomers of TDP-43 CTFs form amyloid fibrils? We showed that the cytoplasmic condensates of GRR (C263 and C274) were quite rarely formed but those of C274 contained amyloid fibrils (Supplementary Fig. 2). The GRR (C274) was incorporated into the condensation of F220 and also into that of C220Δ (Figs. 4, 6). Since C220Δ shares the region of F220 despite lacking the HR, the condensates were formed similarly to F220 (Fig. 5). Optogenetic induction of intermolecular assembly using CRY2olig showed that the N-terminal end of GRR could be closed intermolecularly, while that of TDP25/C220 could not. On the other hand, the N-terminus of GRR (C263) allows it to be proximate, suggesting that, before the conformational changes of GRR affected by such as tRRM2, GRR structure may be flexible and is likely to accept to interact with other molecules depending on its conformation. Since the C-terminus of GRR is difficult to proximate (Fig. 8), the molecular assembly of GRR on the N-terminal side can lead to its conformational change, and then the amyloidogenic cross-β sheet transition can progress. tRRM2 may work not only as a driving force for intermolecular assembly of TDP-43 CTFs but also as an opportunity to change the structural transition of GRR, resulting in condensation including amyloid structures with progressive propagation involving the other chains of TDP-43 CTFs having non-amyloid conformations (Supplementary Fig. 11b). Furthermore, there may be some limiting conditions that

predispose to the transformation of a disordered GRR conformation into amyloid fibrils in the neurodegenerative disorders: cleavage and unusual translation of TDP-43 to generate its CTFs that should be misfolded, accumulation of misfolded proteins.

Regarding the role of HR and α-helix in the GRR, since the intermolecular assembly of C263 from the C-terminal side occurred more when HR was lacking (Fig. 7), HR may play a role in maintaining the IDR conformation to inhibit intermolecular proximity; conversely, the helix would undergo a conformational change with a rate-determining step, rather than immediately transitioning to amyloid structures just after the intermolecular proximity. The chimera protein composed of F220 and the C-terminus of TDP-1 of *C. elegans* (F220c) exhibited a similar tendency for the formation of the cytoplasmic condensate formation as F220 (Fig. 5). However, a CTF of TDP-1 (TDP1-C295), corresponding to the human C220 sequence, demonstrated toxic condensation properties similar to C220 (Supplementary Fig. 5). This suggests that the combination of the truncated RRM2 and the evolutionarily conserved C-terminal disordered region may play a crucial role in the condensation of TDP-43 and its orthologue CTFs within homologous species. Furthermore, the TDP-43 GRR might assume a role in forming protein complexes through diverse interactions with other proteins (e.g., hnRNP A1/A2 and FUS/TLS), in addition to self-interactions of TDP-43 during the evolution process to higher organisms[31].

However, another question arises here. As reported in the analysis of the pathological amyloid structure of TDP-43 described above, the core structure of the amyloid fibrils of TDP-43 consists of only polypeptide chains of the GRR. Our analysis using FCS suggests that, on average, many TDP25/C220 form heterogeneous oligomers with endogenous proteins in the soluble fraction of the cell lysates, but the existence of homo-oligomers is not denied because highly bright molecules, likely corresponding to highly homogeneous oligomers, were observed during measurements (Fig. 2a). The highly homogeneous oligomers would primarily accumulate as

amyloid fibrils, resulting in amyloid fibrils-containing condensates, whereas the heterogeneous oligomers including various endogenous proteins would remain in the cytoplasm with the maintenance of the diffusing state. As a result, the heterogeneous oligomers of TDP25/C220 can lead to loss-of-function of various endogenous proteins that are incorporated into those oligomers and then would dysregulate cellular proteostasis. On the other hand, since there are many reports to show the toxicity of amyloid fibrils[66,67], diffuse homogenous oligomers including amyloid fibrils would also produce cytotoxicity not by loss-of-function but, for example, gain-of-function manner.

C233 formed homogenous oligomers the most efficiently among TDP43 CTFs that were examined here (Figs. 2 and 9d). Since C233 lacks more than half of β3 strands, β5 may form a metastable structure with β4, which could allow homo-oligomerization of C233, not likely to prefer the hetero-oligomerization with endogenous proteins (Supplementary Fig. 11c). Since the condensation efficiency of C253, which only has β5 as tRRM2, was higher than that of C233 and C243, which includes also β4, the β5 probably becomes the conformation unstable (Supplementary Fig. 11d, e). Since β4- and β5-linked peptide (246–258) forms amyloid-like fibrils, while the peptide containing α2 (233–245) did not[39], the region containing α2 can prevent structural changes and misfolding of tRRM2 (Supplementary Fig. 11c). C233 includes α2, and the side chain of Gln 241 in α2 and Asn 259 in β5 forms electrostatic interaction between amine and carbonyl groups in intact RRM2 observed in a determined structure (PDB#3d2w). Since this electrostatic interaction between Gln 241 and Asn 259 exists near the tip of the U-shaped chain formed by α2, β4, and β5 (Supplementary Fig. 12), this electrostatic interaction may suppress the dissociation of β4 and β5, likely contributing to the formation of stable homogenous oligomers. Furthermore, such a stable conformation of the C233 oligomers may be a possible reason why C233 formed fewer condensates in Neuro2a cells. Such stable C233 oligomers are diffusive, but less likely to interact with other endogenous proteins, resulting in low toxicity compared to TDP25/C220. Consequently, stabilization of misfolded tRRM2 is important to prevent the production of toxic oligomers; it is thus considered to be a major target for controlling and preventing the aggregation of TDP43 CTFs containing tRRM2. A cytoplasmic molecular chaperone, 70 kDa heat shock protein (HSP70) prevents TDP25/C220 condensation through the direct interaction between them[68,69]. C220 can be purified as a soluble protein from Neuro2a cell lysate when it is bound to HSP70[58]. HSP70 can recognize and stabilize the conformations of tRRM2 of C220. Furthermore, since HSP70 is entrapped in TDP25/C220 condensates and associates and dissociates with its condensates relatively slowly[69], TDP25/C220 condensates would induce chaperone dysfunction by depriving the cytosol of the functional requirement of HSP70.

In contrast, a striking difference in worms from the Neuro2a cells is that the condensates of C233 were frequently observed as same as that of TDP25/C220 (Fig. 9a), and the fluidity of C233 in the condensates was dramatically higher than TDP25/C220 (Fig. 9c). The soluble oligomers of C233 abundantly contain homogeneous ones, which is similar to Neuro2a cells (Fig. 9d, e). Homogeneous oligomers of C233 would dynamically move between the cytoplasm and the condensates. Protein condensates are formed as a result of stress such as heat shock and aging[70,71]. The biology of condensates is likely various because the condensates formed in response to acute stress such as heat shock are reversible[72]; however, these processes could be perturbed in the presence of disease proteins and/or during ageing[73]. Thus, there would be a mechanism that makes condensates more likely in the body-wall muscle in worms compared to Neuro2a cells. Proteome stability and proteostasis maintenance are widely known to be crucial for the cellular function and lifespan of organisms, and it is important for the maintenance of muscle cells and neurons[42,74]. These findings suggest that misfolded protein oligomers such as C220/TDP25 and C233 and their condensates composed of these oligomers would be toxic and make damage the body-wall muscles. Consequently, we speculate that the cytotoxicity in the body-wall muscles would be involved not in the condensates but the characteristics of the diffuse oligomers (e.g., heterogeneity). Furthermore,

the cell-to-cell transmission of neurodegenerative disease-associated protein oligomers, such as TDP-43 and ALS-associated and aggregation-prone dipeptide repeat peptides transcribed from the *C9orf72* locus, has been demonstrated[75,76]. In addition to these reports, one may speculate that dynamic oligomers are more easily transmitted among the cells than static condensates. In the worms we analyzed, the fluorescence molecules were localized only in the body-wall muscles, so that such cell-to-cell transmission of the oligomers would be in circumscribed organs; however, since the condensates were distributed in the body-wall muscles of GFP-positive areas, such transmission may contribute to the prominent form of the condensates especially in C233 in the body-wall muscles in worms compared to Neuro2a.

The worms expressing C220 and C233 in the body-wall muscles showed a shortening of their lifespan (Fig. 9g, h). A worm model that expresses polyglutamine-expanded mutant huntingtin, which causes the inherited neurodegenerative disorder Huntington's disease, in the body-wall muscles showed a significantly shorter lifespan[46]; however, expression of polyglutamine-expanded Ataxin-3 in the body-wall muscles did not affect the lifespan in this model[77]. Although it is still unknown why such differences in the lifespan when the aggregation-prone proteins are expressed in the body-wall muscles, a possible explanation is that the expression level of such proteins would be relevant. Furthermore, the shortening lifespan is involved in the insulin/insulin-like growth factor-1 signaling (IIS) pathway, a major and well-conserved pathway that is involved in the longevity and the proteostasis maintenance in *C. elegans* across eukaryotes[42,78]. The key regulator of IIS in *C. elegans* is DAF-16, a FOXO transcription factor homolog[79], The major body-wall muscles health regulator Axin increased longevity through the AMPK signaling pathway and DAF-16 in the IIS[80]. The short lifespan of the worms expressing C220 and C233 in the body-wall muscles is likely involved in some defections in such signaling pathways maintaining longevity. As elevated oxidative stress exacerbated the lifespan difference between worms expressing C220 and C233 (Fig. 9h), it supports the contribution of the IIS pathway to the toxic manifestation of TDP-43 CTFs in the body-wall muscles. The worms pan-neuronally expressing TDP25 showed neurotoxicity[45]; however, their lifespan has not been elucidated. Therefore, the worms that express C220 and C233, as we established, should be a useful model that shows the phenotype reflecting the aggregation characteristics and cytotoxicity of TDP-43 CTFs, and for analyzing proteostasis dysregulation by misfolded protein and IDPs.

We characterized the aggregation mechanism and toxicity of TDP-43 CTFs using immortalized cell lines and worms. Misfolded tRRM2 may promote intramolecular assembly of not only TDP-43 CTFs but also intact TDP-43 as a seed of aggregation (Fig. 10a). Destabilized β-strands in tRRM2 may involve the helix-beta transition of the prion-like domain in GRR, resulting in a fibril core formation of amyloid-like aggregates (Fig. 10b). The model system that we established in this study is considered to be valuable as an experimental tool for elucidating pathological TDP-43 aggregation and its inhibition mechanisms in the future.

## Methods
### Construction of mammalian expression plasmids
To create expression plasmids for Neuro2a cells, DNA fragments of C220, C233, C243, C253, C263, and C274, were amplified by PCR using the following primers (Hereafter, underlines of the primer sequences denote restriction enzyme sites for cloning): 5′-GTCAAGCTTCCACCATGGTCTTCATCCCCAAGCCATTCAGG-3′ (forward side for C220), 5′-GCTAAGCTTCGACATTTGCAGATGATCAGAT-3′ (forward side for C233), 5′-GCTAAGCTTCGCTTTGTGGAGAGGACTTGAT-3′ (forward side for C243), 5′-GCTAAGCTTCGATCAGCGTTCATATATCCAA-3′ (forward side for C253), 5′-GCTAAGCTTCGAAGCACAATAGCAATAGAC-3′ (forward side for C263), 5′-GCTAAGCTTCGGGAAGATTTGGTGGTAATC-3′ (forward side for C274), and 5′-CGTGGGATCCCTAGTGATTCATTCCCCAGCCAG-f3′ (reverse side for all of TDP-43 CTFs including glycine-rich region). PCR products were digested and ligated into the HindIII and BamHI sites of pmeGFP-C1[23] containing cDNA for a

monomeric variant of enhanced GFP that contains the mutation A206K (in this paper, it is named GFP) (pGFP-C220, -C233, -C243, -C253, -C263, and -C274). DNA fragments of F220 (220–262) and RRM2 (190–262) were amplified by PCR using the following primers: 5′-GCTAAGCTTCGAGAA AAGTGTTTGTGGGGCGCTG-3′ (forward side of RRM2), the same one for the forward side of C220 (forward side of F220), and 5′-CG TGGATCCTAAGGTTCGGCATTGGATATA-3′ (reverse side of F220 and RRM2). These fragments were inserted into the HindIII and BamHI sites of pmeGFP-C1 as described above (pGFP-F220 and -RRM2). To make a domain-swap mutant (SW), in which F220 was fused to the C-terminal of GRR, two DNA fragments were amplified by PCR using the following primers: 5′-GCTAAGCTTCGGGAAGATTTGGTGGTAATC-3′ (Forward side for GRR of SW), 5′-CAGGTCGACCATTCCCCAGCCAGAAG ACTT-3′ (Reverse side for GRR of SW), 5′-CAGGTCGACGTCTTCA TCCCCAAGCCATTC-3′ (Forward side for F220 of SW), and the same one for the reverse side of F220 and RRM2 (Reverse side for F220 of SW). These fragments were ligated via the SalI site and inserted into the HindIII and BamHI sites of pmeGFP-C1 as described above (pGFP-SW). To make an expression plasmid for C220Δ (220-414 lacking 311-341 of TDP-43), two DNA fragments were amplified by PCR using the following primers: the same one for the forward side of C220 (forward side of N-terminal fragment of C220Δ), 5′-CCCACCACCCATATTACTAC-3′ (reverse side of N-terminal fragment of C220Δ), 5′-GCCAGCCAGCAGAACCAGTC-3′ (forward side of C-terminal fragment of C220Δ), and the same one for the reverse side of C220 (reverse side of C-terminal fragment of C220Δ). These fragments were phosphorylated by T4 polynucleotide kinase, and then directly inserted into pmeGFP-C1 (pGFP-C220Δ). To make an expression plasmid for C274Δ (274–414 lacking 311–341 of TDP-43), a DNA fragment was amplified by PCR using the following primers: 5′-GCT AAGCTTCGGGAAGATTTGGTGGTAATC-3′ (forward side of C274Δ) and 5′-CGTGGGATCCCTAGTGATTCATTCCCCAGCCAG-3′ (reverse side of C274Δ). The fragment was inserted into the HindIII and BamHI sites of pmeGFP-C1 as described above (pGFP-C274Δ). To prepare the plasmids for expression of TDP-43 CTFs tagged with an red fluorescent protein mCherry (RFP), required fragments for TDP-43 CTFs were obtained by digestion of plasmids described above at the HindIII and BamHI sites, and then inserted into pmCherry-C1 (pRFP-C220, -C274, and -C274Δ).

The cDNA fragments of TDP43Δ (TDP-43 lacking 311-341 amino acids and carrying a mutation of nuclear localization signal sequence) and TDP43CSΔ (TDP43Δ carrying C173/175S and nuclear localization signal sequence mutations) were ordered in gene synthesis services (Eurofins Genomics Inc., Tokyo, Japan), and then inserted into pmeGFP-N1 or pmCherry-N1.

The synthetic DNA fragment for a chimera encoding amino acids 220–262 of human TDP-43 fused to the C-terminal region of TDP-1 (F44G4.4), a TDP-43 orthologue in *C. elegans* (GTCAAGCTTCGGTCTTC ATCCCCAAGCCATTCAGGGCCTTTGCCTTTGTTACATTTGCAGAT GATCAGATTGCGCAGTCTCTTTGTGGATTAAAGGAATCAGCGGA GGACTTGATCATTCATATATCCAATGCCGAACCTAAGCACAATA GCAATAGATCAGTTGGCCCTGATTATGGCCTTCCAGCTGGCTAC CGTAACCGCAGAGAACGTGATCGACCGGATAGACGACCGATTCA AAATGAAGCACCTCTGCCCATGCCATTCGTCCGCCCACCACAAG ATTACTCATACCGTCAGCAAAATTCTCCCCTCGAGAGGAGATAC TGGGCACCTGGAGACTCGAGAGGACCAGGATGGCGGGATCCG TC) was ordered in GeneArt gene synthesis services (Thermo Fisher, Waltham, MA, USA). The fragment was digested and inserted into the HindIII and BamHI sites of pmeGFP-C1 (pGFP-F220c). The DNA fragment for 295–411 of TDP-1 was amplified by PCR from plasmids containing TDP-1 cDNA sequence using the following primers: 5′-GCAAAGCTTCGG TGTTCATCCCAAAACCATTC-3′ (forward side) and 5′-CGAGGAT CCTCACCATCCTGGTCCTCTCGAG-3′ (reverse side). The fragment was digested and inserted into pmeGFP-C1 at the HindIII and BamHI sites.

To make plasmids for a homo-oligomerization variant of cryptochrome 2 (CRY2olig)-based oligomerization analysis, DNA fragments for N-terminally CRY2olig-tagged TDP-43 CTFs were amplified by PCR from

plasmids containing required sequence as described above using the following primers: 5′-GCTCCCGGGCGGTCTTCATCCCCAAGCC-3′ (forward side of C220 and C220Δ), 5′-GCTCCCGGGCGAAGCACAATA GCAATAGAC-3′ (forward side of C263 and C263Δ), and 5′-CGT ACCGGTGGGTGATTCATTCCCCAGCCA-3′ (reverse side of C220, C220Δ, C263, and C263Δ). The fragments were digested and inserted into mCherry-N1 at the SmaI and AgeI sites; then a fragment of CRY2olig region that was digested from pCRY2olig-mCherry (#60032; Addgene, Watertown, MA, USA) was inserted at the NheI and SmaI sites (pCRY-C220-, -C220Δ-, -C263-, and -C263Δ-R). DNA fragments for C-terminally CRY2olig-tagged TDP-43 CTFs were amplified by PCR from plasmids containing the required sequence as described above using the following primers: 5′- GCAGCTAGCCACCATGGTCTTCATCCCCAAGCCAT-3′ (forward side of C220 and C220Δ), 5′- GCAGCTAGCCACCATGAAGCA CAATAGCAATAGAC-3′ (forward side of C263 and C263Δ), and 5′- CGT CTCGAGCATTCCCCAGCCAGAAGACTTAG-3′ (reverse side of C220, C220Δ, C263, and C263Δ). The fragments were digested and inserted into pCRY2olig-mCherry at the NheI and XhoI sites (pC220-, C220Δ-, C263-, and C263Δ-CRY-R). DNA fragments for C-terminally CRY2olig-tagged meGFP were amplified by PCR from meGFP-C1 using the following primers: 5′- GCAGCTAGCCACCATGGTGAGCAAGGGCGAGGAG-3′ (forward side) and 5′- CGTCTCGAGCTTGTACAGCTCGTCCAT GCCG-3′ (reverse side); then inserted into pCRY2olig-mCherry (Addgene) at the NheI and XhoI sites (pGFP-CRY-R).

## Cell culture and transfection

Murine neuroblastoma Neuro-2a cells (#CCL-131; ATCC, Manassas, VA, USA) and maintained in DMEM (D5796; Sigma-Aldrich, St. Louis, MO, USA) supplemented with 10% FBS (12676029, Thermo Fisher), 100 U/mL penicillin G (Sigma-Aldrich), and 0.1 mg/ml streptomycin (Sigma-Aldrich)[23]. Cells ($2.0 \times 10^5$) were spread in a 35 mm glass-bottom dish for microscopic experiments (3910-035; IWAKI-AGC Technoglass, Shizuoka, Japan) or a 35 mm plastic dish for cell lysis (150318; Thermo Fisher) at 1 day before transfection. The plasmid DNAs (1 μg for expression of TDP-43 CTFs; and the mixture of 0.1 μg pmeGFP-N1 and 0.9 μg pCAGGS for expression of GFP monomers as a control; the mixture of 1.0 μg GFP-tagged TDP-43 CTFs and 1.0 μg RFP-tagged TDP-43 for two color co-expression) were transfected using Lipofectamine 2000 (Thermo Fisher). After 1 overnight incubation for transfection, the medium was exchanged for the flesh and maintained prior to microscopic observation. Before the cell lysis, the medium including the transfection reagent was removed, and the cells were washed in PBS at 25 °C.

## Cell lysis and solubility assay

Neuro2a cells were lysed in a lysis buffer containing 25 mM HEPES-KOH (pH 7.5), 150 mM NaCl, 1% Noidet P-40, 1% sodium deoxycholate, 0.1% SDS, 0.01 U/μL benzonase, and 1× protease inhibitor cocktail (Sigma-Aldrich). After the centrifugation (20,400 *g*, 20 min., 4 °C), the supernatants were recovered, and the pellets were solubilized in PBS containing 1 M urea. SDS-PAGE sample buffer-mixed lysates were denatured at 98 °C for 5 min. Samples were applied to a 5–20% polyacrylamide gel (ATTO, Tokyo, Japan) and subjected to electrophoresis in an SDS-containing buffer. The proteins were transferred to a PVDF membrane (Cytiva, Marlborough, MA). The membrane was incubated in 5% skim milk in PBS-T for background blocking for 1 h. After incubation with an anti-GFP antibody (#GF200, Nacalai tesque, Kyoto, Japan), an anti-mCherry antibody (#Z2496N, TaKaRa), or a horseradish peroxidase-conjugated anti-α-tubulin antibody (#HRP-66031, ProteinTech, Rosemont, IL, USA) in CanGet signal immunoreaction enhancer solution 1 (TOYOBO, Osaka, Japan), horseradish peroxidase-conjugated secondary antibodies that are appropriate for primary antibodies (#111-035-144 or #115-035-062; Jackson Immuno Research, West Grove, PA, USA) were incubated in 5% skim milk in PBS-T. Chemiluminescence signals were acquired using a ChemiDoc MP imager (Bio-Rad, Hercules, CA, USA). The intensities of the bands were measured using Fiji-ImageJ. The insoluble proportion was normalized as the intensity

of the band in the pellet was divided by the sum of the intensities band both in the supernatant and pellet.

For filter retardation (FR) assay, cells were lysed in a PBS buffer containing 1% SDS and 0.25 U/mL of benzonase (Sigma-Aldrich). The lysates were applied to a cellulose acetate membrane with a pore size of 0.2 μm (Advantec Toyo, Tokyo, Japan) using a Bio-Dot SF blotter (Bio-Rad). The membrane was blocked in PBS-T buffer containing 5% skim milk for 1 h. The GFP on the membrane was detected by western blotting, as described above. All the intensities were measured using Fiji-ImageJ.

### Confocal fluorescence microscopy of Neuro2a cells
Neuro2a cells were observed using an LSM 510 META (Carl Zeiss, Jena, Germany) with a C-Apochromat 40×/1.2NA W Korr. UV-VIS-IR water immersion objective (Carl Zeiss). GFP and mCherry were excited at 488 nm and 543 nm, respectively. Excitation beams were divided through HFT488/543. Fluorescence was divided through a dichroic mirror (NFT545) and corrected through a 505–530 nm band-pass filter (BP505-530) for GFP and a 585 nm long-pass filter (LP585) for mCherry. Pinhole was set to 71 and 82 μm.

### Counting of condensate-positive and dead cells
Neuro2A cells expressing TDP-43 CTFs tagged with GFP were prepared as described in the other experiments. The number of condensate-positive cells was counted by observing various fluorescence intensity images of the same field of view and then divided by the number of GFP-positive cells. For dead cells, on 1 day after the transfection, dead cells were stained with a 1.0 μg/mL propidium iodide (PI) solution (Thermo Fisher Scientific) for 5 min. Images of the GFP and PI channels were acquired using an LSM 510 META microscope through a Plan-Neofluar 10×/0.3NA objective at 37 °C. The percentage of dead cells was calculated from the number of PI-positive cells divided by the number of GFP-positive cells. Cell spreading, DNA transfection, and fluorescence observation were conducted as one trial, and multiple independent experiments ('n' in figure legends) were performed.

### Amytracker staining
Neuro2a cells expressing TDP-43 CTFs tagged with GFP were fixed in 4% paraformaldehyde buffered with 100 mM Hepes-KOH (pH 7.5). After washing in tris-buffered saline (TBS), cells were stained with a far-red amyloid-specific fluorescent dye (Amytracker 680; Ebba Biotech, Stockholm, Sweden) diluted in PBS (1/500) for 90 min. The excess dye was washed in PBS. Fluorescence images were acquired using an LSM 510 META (Carl Zeiss) and a C-Apochromat 40×/1.2NA W Korr. UV-VIS-IR M27 water immersion objective lens (Carl Zeiss).

### Hexanediol treatment of Neuro2a cells
Neuro2A cells expressing TDP-43 CTFs tagged with GFP were prepared as described in the other experiments and grown on a 35 mm glass base dish (IWAKI-AGC Technoglass). On 1 day after the transfection, the medium was exchanged for flesh and maintained before microscopic observation; then, the cells were incubated for 1 h. Cells were observed using an LSM 510 META (Carl Zeiss) with a C-Apochromat 40×/1.2NA W Korr UV-VIS-IR M27 water immersion objective. Cells were treated with 6% (w/v) 1,6-Hexanediol (1,6-HD) (Tokyo Chemical Industry, Tokyo, Japan) on the microscope stage at 37 °C with 5% CO$_2$. Confocal imaging of cells positive for the condensates was performed as described in the method for confocal microscopy. Z-stack images were acquired for 7 slices at 2.0 μm intervals. Time-lapse observation of 1,6-HD treated cells was performed until the cells were shrunk or swelled at 5 min. intervals (approx. for 25 min).

### Blue-light induced oligomerization assay in live cells
Plasmids for the expression of CRY2olig-tagged proteins (pCRY2olig-R, pGFP-CRY-R, pCRY-C220-R, pCRY-C220Δ-R, pCRY-C263-R, pCRY-C263Δ-R, pC220-CRY-R, pC220Δ-CRY-R, pC263-CRY-R, and pC263Δ-CRY-R) were transfected into Neuro2a cells at 1 day before the experiments. After incubation for 24 h, the medium was exchanged for a normal growth medium. Cell selection for observation, blue-light irradiation, and imaging was performed using an inverted fluorescence microscope, Axioobserver Z1 (Carl Zeiss) combined with an LSM 510META (Carl Zeiss) and a C-Apochromat 40x/1.2NA W Korr. UV-VIS-IR water immersion objective. Cells expressing CRY2olig-tagged proteins were selected by ocular observation using fluorescence filter sets for mCherry (#49008, Chroma, Bellows Falls, VT, USA) with a mercury lamp. Light irradiation for oligomerization and observation was performed using the LSM. mCherry was excited at 594 nm. The excitation and emission lights were split using a beam splitter (NFT488/594). Fluorescence was collected through a 615 nm long-pass filter (LP615) and through a 545 nm dichroic mirror (NFT545). The pinhole diameter was 150 μm. After 1 frame observation, cells were irradiated at 488 nm (7.04 ± 0.2 μW; mean ± SD; $n = 3$ trials); pixel dwell time: 0.64 μs; 3 iterations. After 488 nm irradiation, cells were observed at 20 s intervals. Cells forming foci within 2 min were counted. A hypothesis test for the difference in the population proportions was performed using a BellCurve for Excel software (Social Survey Research Information Co., Ltd., Tokyo, Japan).

To observe amyloids, cells were stained with a green amyloid-specific fluorescent dye (Amytracker 520; Ebba Biotech) diluted in Opti-MEM I (Thermo Fisher) (1/500) for 30 min. Fluorescence images were acquired using an LSM 510 META (Carl Zeiss) and a C-Apochromat 40×/1.2NA W Korr. UV-VIS-IR M27 water immersion objective lens (Carl Zeiss).

### Construction of worm expression plasmids
For the nematode strain, DNA fragments of meGFP-tagged C220 and C233 were amplified from plasmids pGFP-C220 and -C233, respectively, by PCR using the following primers: 5′-CTCGCTAGCAAAATGGTGAGCAAG GGCGAGGAGC-3′ (forward side) and 5′-GAGGGTACCTTAGTGATTC ATTCCCCAGCCAGAAGAC-3′ (Reverse side). A DNA fragment of meGFP monomers was amplified by PCR using the following primers: 5′-C TCGCTAGCAAAATGGTGAGCAAGGGCGAGGAGC-3′ (forward side) and 5′- GAGGGTACCTTACTTGTACAGCTCGTCCATGCCGAG-3′ (reverse side). PCR products were digested and ligated into the NheI and KpnI sites of pPD30.38 containing the *myo-3* promoter and enhancer elements from the *unc-54* myosin heavy-chain locus (pPD30.38::meGFP-C220; pPD30.38::meGFP-C233; pPD30.38::meGFP)[48].

### C. elegans culture and synchronization
Nematodes were handled using standard methods and cultured at 20 °C[81]. For the generation of transgenic animals, 100 ng/μl plasmid DNA encoding meGFP-tagged TDP-43 CTFs (pPD30.38::meGFP-C220, pPD30.38::meGFP-C233, and pPD30.38::meGFP) were linearized with PvuII and mixed with the same volume of 100 ng/μl pBluescript. The mixtures were microinjected into the gonads of adult hermaphrodite N2 animals (Caenorhabditis Genetics Center, Minneapolis, MN, USA). Transgenic animals containing extrachromosal arrays were selected and isolated as strains based on fluorescence in body-wall muscle cells (C220, C233, and GFP strains).

For nematode assays, eggs were recovered by treating young adult animals with an alkaline hypochlorite solution containing 0.25 M NaOH and 20% NaOCl (effective chlorine concentration ~ 1%). The eggs were hatched overnight by gently shaking them in M9 buffer at 20 °C. The next day, the hatched larvae (1 day after synchronization by bleaching) were grown at 20 °C on nematode growth medium (NGM) agar plates seeded with OP50 *E. coli* (Caenorhabditis Genetics Center) as a food source. GFP fluorescence-positive nematodes (3 days after synchronization by bleaching) were selected using a stereo-dissection microscope Stemi 305 Trino (Carl Zeiss) with an epi-illumination 470 nm light source (LED470MS-EPI; Optocode Corp., Tokyo, Japan), a 495 nm long passing glass filter (FGL495S; Thorlabs, Newton, NJ, USA), and a 550 nm short passing glass filter (SV0550; Asahi Spectra, Tokyo, Japan), and were transferred to new NGM plates seeded with OP50 using a nematode-picker (platinum wire). Until the following assays, the nematodes were maintained on NGM agar plates seeded with OP50 at 20 °C and were transfered to new plates every day.

## Confocal fluorescence microscopy of C. elegans

The nematodes were placed on a 1% agarose pad with one drop of 8 mM Levamisole (Tokyo Chemical Industry) and mounted in Halocarbon oil 700 (Sigma-Aldrich) with a No. 1S coverglass (Matsunami, Osaka, Japan). Mounted nematodes were observed using an LSM 510 (Carl Zeiss) with a Plan-Neofluar 10×/0.3NA objective (Carl Zeiss). GFP was excited at 488 nm. The excitation beam was divided through HFT 488. Fluorescence was selected through a 505-550 nm band-pass filter (BP505-550). Bright-field images were acquired using a transmitted light detector for LSM 510 in an inverted microscope Axiovert 100 M (Carl Zeiss). The pinhole was set to 140 μm, The X- and Y-scanning sizes were each 1024 pixels, the zoom factor was set to 0.8×, and the pixel dwell time was 3.2 μs.

## C. elegans lysis and western blotting

Synchronized nematodes expressing GFP monomers, G-C220, or G-C233 ($n = 200$) were corrected at 4 or 8 days after the synchronization by bleaching (day 1 or 5 adulthood) and washed in PBS at 20 °C 3 times. After the nematodes were suspended in a lysis buffer containing 20 mM Hepes-KOH (pH 8.0), 420 mM NaCl, 0.2 mM EDTA, 25% glycerol, 0.5 mM DTT, and 1× protease inhibitor cocktail (Sigma-Aldrich), nematodes were frozen at −80 °C and thawed for 3 times. After the addition of 1% SDS and 1 M urea, the suspensions were sonicated, then the supernatant was recovered after the centrifugation (20,400 $g$, 20 min., 25 °C). After addition of SDS-PAGE sample buffer, the lysates were denatured at 98 °C. The samples were applied to a 12.5% polyacrylamide gel and subjected to electrophoresis in an SDS-containing buffer. The proteins were transferred to a PVDF membrane (Cytiva). The membrane was incubated in 5% skim milk in PBS-T for background blocking for 1 h. After incubation with an anti-GFP antibody (#GF200, Nacalai tesque) or anti-α-tubulin antibody (#sc-32293, Santa Cruz Biotechbology, Dallas, TX, USA) in CanGet signal immunoreaction enhancer solution 1 (TOYOBO), an alkaline-phosphatase-conjugated goat anti-mouse IgG secondary antibody (#SA00002-1, ProteinTech) were incubated in 5% skim milk in PBS-T. Bands were stained in 5-bromo-4-chloro-3-indolyl-phosphate and nitro blue tetrazolium solution (Sigma-Aldrich). The membrane image was acquired using a digital steel camera EOS M6 mark II with a lens EF16-35 mm F4L IS USM (Canon Inc., Tokyo, Japan).

## Motility analysis

To analyze the number of body bends, five or fewer nematodes at 4 or 8 days after the synchronization by bleaching were transferred with a worm-picker from growing NGM agar plates to bacteria-lacking ones to clean nematodes from bacteria. M9 buffer (1 ml) was gently poured from the edge into the NGM agar plates. After the incubation for 1 min, the bending worms were recorded for 30 s using a digital steel camera EOS kiss digital X3 (Canon Inc.) attached to a stereo-dissection microscope Stemi 305 Trino (Carl Zeiss). The $x$- and $y$-image sizes of the movie were 1920 and 1080 pixels, respectively; and the frame rate was 20 fps. Repeatedly, the bending worms were recorded, and movies for 36 animals were obtained. One body bend corresponds to the movement of the head region that thrashes from one side to the other and back to the starting position. The number of bends was counted by observation of the recorded movie.

## Lifespan analysis

Synchronized nematodes (3 days after bleaching; L4 larvae) were transferred onto NGM agar plates seeded with OP50 E. coli and maintained at 20 °C. The next day, each nematodes at 4 days after the synchronization (Day 1 of adulthood) were transferred to independent NGM agar plates seeded with OP50 after determining the viability of the worms using a stereo dissection microscope Stemi 305 Trino (Carl Zeiss). Worms that did not move when gently prodded and did not show pharyngeal pumping were marked as dead. Two independent trials with 30 worms were conducted. The percentage alive was calculated using all 60 worms of the two independent trials. For oxidative stress assay, each nematodes at 4 days after the synchronization (Day 1 of adulthood) were transferred to NGM agar plates containing

270 μM Juglone (Selleck Chemicals, Houston, TX, USA) and seeded with OP50 after determining the viability of the worms using a stereo dissection microscope Stemi 305 Trino (Carl Zeiss). The generalized Wilcoxon test (Gehan-Breslow method) was performed using a BellCurve for Excel software (Social Survey Research Information Co., Ltd., Tokyo, Japan).

## FRAP

Photobleaching experiments were performed on an LSM 510 META system through a C-Apochromat 40×/1.2NA W Korr. UV-VIS-IR M27 objective lens (Carl Zeiss). GFP was excited and photobleached at 488 nm, and fluorescence was separated using a dichroic mirror (NFT545). GFP fluorescence was collected through a band-pass filter (BP505-570). The pinhole size was set to 200 μm, the zoom factor was set to 5×, and the interval time for image acquisition was set to 10 s. The X- and Y-scanning sizes were each 512 pixels, and the GFP was photobleached at 488 nm, 100% transmission, for less than 1.92 s. The relative fluorescence intensity (RFI) at time point $t$ (RFI($t$)) was measured using Fiji-ImageJ and ZEN (Carl Zeiss) and then calculated as the following equation (Eq. 1).

$$\mathrm{RFI}(t) = \frac{I_{\mathrm{BL}}(t) \cdot I'_{\mathrm{Ref}}}{I'_{\mathrm{BL}} \cdot I_{\mathrm{Ref}}(t)} \tag{1}$$

where $I_{\mathrm{BL}}(t)$ and $I_{\mathrm{Ref}}(t)$ are the intensity at time point $t$ in the photobleached region and the reference region, respectively. $I'_{\mathrm{BL}}$ and $I'_{\mathrm{Ref}}$ are the intensity before photobleaching. To obtain the photobleaching efficiency just after the photobleaching (the lowest fluorescence intensity in the photobleached region), cells were fixed in 4% paraformaldehyde (PFA)-containing 100 mM Hepes-KOH (pH 7.5) buffer for 19 h, followed by washing in TBS, and photobleaching of GFP in the fixed cells was performed under the same condition ($n = 10$). Normalized relative fluorescence intensities (nRFI) were calculated by subtracting the mean photobleaching efficiency from the RFI. To calculate the maximum recovery proportion, recovery curves associated with the relative fluorescence intensity were fitted using a single-component exponential recovery model in Origin 2024 software (OriginLab Corp., Northampton, MA, USA).

The nematodes were placed on a 1% agarose pad with one drop of 8 mM Levamisole (Tokyo Chemical Industry) and mounted in Halocarbon oil 700 (Sigma-Aldrich) with a No. 1 S coverglass (Matsunami). Mounted nematodes were observed using an LSM 510 (Carl Zeiss) with a C-Apochromat 40×/1.2NA Korr. water immersion objective (Carl Zeiss). GFP was excited at 488 nm. The excitation beam was divided through HFT 488. Fluorescence was selected through a 505-550 nm band-pass filter (BP505-550). The pinhole was set to 300 μm, and the zoom factor was set to 15×. X- and Y-scanning sizes were 512 and 256 pixels, respectively. GFP was photobleached at 488 nm, 100% transmission, for less than 1.68 s. After the photobleaching, images were acquired at 15 s intervals. The nRFI was obtained by the photobleaching experiment using methanol-fixed nematodes ($n = 10$) as described for Neuro2a cells.

## FCS

Neuro2a cells transiently expressing GFP-tagged TDP-43 CTFs were lysed in a buffer consisting of 50 mM Hepes-KOH (pH 7.5), 150 mM NaCl, 1% Noidet P-40, and 1× protease inhibitor cocktail (Sigma-Aldrich). Supernatants of the lysates were recovered after centrifugation at 20,400 $g$ for 10 min. at 4 °C, and immediately followed by FCS measurement. For nematodes, in day 4 or 8 after the synchronization, nematodes ($n = 200$) were collected and shaken in PBS for 1 h at 20 °C to remove E. coli. Nematodes were suspended in a lysis buffer containing 100 mM Tris-HCl (pH 7.4), 10 mM MgCl$_2$, 1% Triton X-100, 1× protease inhibitor cocktail (Sigma-Aldrich). The freeze-thawed nematodes were sonicated using an ultrasonicator. The supernatants were recovered after centrifugation of the sonicated samples at 20,400 $g$ for 1 min at 4 °C and immediately followed by FCS measurement. FCS was performed using an LSM 510 META + ConfoCor2 (Carl Zeiss) with a C-Apochromat 40×/1.2NA W Korr. UV-VIS-IR water immersion objective lens (Carl Zeiss). Photons were recorded

for 300 s. Curve fitting analysis for acquired autocorrelation function was performed using ZEN 2009 software (Carl Zeiss) using a model for one-component 3D diffusion with one-component exponential decay as the following Eq. 2.

$$G(\tau) = 1 + \left[1 + \frac{T}{1-T}\exp\left(-\frac{\tau}{\tau_T}\right)\right]\frac{1}{N}\left(1 + \frac{\tau}{\tau_D}\right)^{-1}\left(1 + \frac{\tau}{s^2\tau_D}\right)^{-\frac{1}{2}}$$

(2)

where, $G(\tau)$ is an autocorrelation function of the time interval $\tau$; $\tau_D$ is the diffusion time; $N$ is the average number of particles in the confocal detection volume; $T$ and $\tau_T$ are the exponential decay fraction and time, respectively; $s$ is the structural parameter determined by measuring the standard fluorescent dye (Alexa Fluor 488) on the same day. Apparent molecular weights were calculated as a Stokes-Einstein relation-modified Eq. 3.

$$Mw_s = \left(\frac{\tau_{D,S}}{\tau_{D,GFP}}\right)^3 Mw_{GFP}$$

(3)

where, $Mw_s$ and $Mw_{GFP}$ (=27 kDa) are the molecular weights of the sample and GFP monomers, respectively; $\tau_{D, S}$ and $\tau_{D, GFP}$ are the diffusion times of the sample and GFP monomers, respectively.

### Statistics and reproducibility
Data obtained from each experiment were expressed as the mean ± 95%CI. The sample sizes were described in each figure legend.

### Data availability
The source data for the main graphs are provided in Supplementary Data 1. Uncropped and unedited blots represented in Figs. 1d, 3d, 5e, 6d, 7f, and 9b were provided in Supplementary Fig. 13.

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

## Acknowledgements

We thank Rintaro Kawamura for insightful discussions and Sue Fox for technical support. The authors acknowledge Open Facility Division, Global Facility Center, Creative Research Institution, and Hokkaido University for the use of a chemiluminescence imager. A.K. was supported by grants from the Japan Agency for Medical Research and Development (JP22gm6410028 and JP22ym0126814); a Japan Society for Promotion of Science (JSPS) Grant-in-Aid for Transformative Research Areas (A) (24H02286); Grant-in-Aid for Scientific Research on Innovative Areas (22H04826); a JSPS Grant-in-Aid for the Promotion of Joint International Research (Fostering Joint International Research) (16KK0156); a JSPS Grant-in-Aid for Scientific Research (C) (18K06201); a grant from Hokkaido University Office for Developing Future Research Leaders (L-Station); a grant from Canon Foundation; a grant from Hoansha Foundation; a grant from Hagiwara Foundation of Japan; a grant from Nakatani Foundation. M.K. was partially supported by a JSPS Grant-in-Aid for Scientific Research (B) (22H02578) and a JSPS Grant-in-Aid for Challenging Research (Exploratory) (22K19886); A.F. was supported by the Support for Pioneering Research Initiated by the Next Generation (SPRING) program by the Japan Science and Technology Agency (JST) in Hokkaido University (JPMJSP2119); and Y.H. was supported by the establishment of universities fellowship toward the creation of science technology innovation (the Hokkaido University Ambitious Doctoral Fellowship) supported by the Japan Science and Technology Agency (JST) (JPMJFS2101).

## Author contributions

Akira Kitamura: Conceptualization; Supervision; Funding acquisition; Investigation; Data curation; Visualization; Methodology; Resources; Writing–original draft; Writing–review and editing. Ai Fujimoto: Investigation; Methodology; Visualization; Funding acquisition; Writing–review and editing. Rei Kawashima: Investigation; Methodology; Visualization. Yidan Lyu: Investigation. Kotestu Sasaki: Investigation. Yuta Hamada: Investigation; Funding acquisition. Kanami Moriya: Investigation; Visualization. Ayumi Kurata: Investigation. Kazuho Takahashi: Investigation. Reneé Brielmann: Investigation; Methodology; Resources. Laura C. Bott: Methodology; Writing–review and editing. Richard I. Morimoto: Supervision; Writing–review and editing. Masataka Kinjo: Funding acquisition; Resources; Writing–review and editing.

## Competing interests

The authors declare no competing interests.

## Inclusion & ethics statement

All collaborators of this study who have met the criteria for authorship required by Nature Portfolio journals have been included as authors, as their participation was essential for the design and implementation of the study. Roles and responsibilities were agreed upon among collaborators before the research.
