## [Peer Review File · Communications Biology]

Reviewers' comments:

Reviewer #1 (Remarks to the Author):

In this work, Kitamura et al have investigated the effect of CTD fragments (CTFs) in the oligomerization of TDP-43. In his work, using both cellular and an animal (c.elegans) model they have investigated the influence of RRM2 misfolding on the amyloidogenic properties of the CTFs. In addition, the authors have taken advantage of their model systems to investigate the proximity properties of the N- and C-terminal regions of TDP-43. From the point of view of the novelty, it has been known since a long time that TDP-43 CTFs promote aggregation and are potentially toxic to cells, with many studies addressing this specific issue. Nonetheless, the work contains some novel data especially due to the use of advanced experimental techniques which have not been employed in the past and some interaction experiments (such as those reported for the C220 and C274 fragments) and that the N- and C-terminals TDP-43 CTFs are close to each other (although the hypothesis that TDP-43 N- and C-terminal interact with each other has already been proposed in the past by the Petrucelli lab several years ago). However, the work also contains several important weaknesses and, as a result, several specific and general issues will need to be addressed to improve the significance of the study:

1) First, the authors should provide some rationale for using the artificial truncation fragments shown in Figure 1A. The reason is that there is a lot of literature where cleavage sites in CTF fragments found in patients/enzymatic cleavage studies have been mapped precisely (ie. 174, 239, and 220/219 for the TDP25 fragment and 208, 247, 274, 291, 306 and 314 for lower size fragments. Would it not have been better to use all this information to plan for the CTF fragments to be used in this project? Second, in Figure 1 the authors should also add results for the C274 fragment for all experiments shown in this figure because this fragment is only used from Figure 2 onwards and is a key component for Figure 4 experiments.

2) Figure 2C is truncated and cannot be seen.

3) Although they have been reported in Figure 1B, in Figure 3B the authors should also include images for C220 and C263 (in the current version only F220 RRM2 and SW are present). This is important because in the quantification of Figure 3C both C220 and C263 are present, and it is not clear if these are new experiments, or they refer to the data reported in Fig.1B.

4) There is also an important discrepancy in the results for C220 in Figures 1D, Figures 3D/5E, and Figure 6F. In Figure 1D, the distribution of C220 in the S and P fraction is heavily predominant in the P fraction, in Figures 3D and 5E the distribution seems to be equal in

the S and D fractions, and in Figure 6F the C220 fragment is present predominantly in the S fraction. All these huge differences raise the question of the variability observed in these experiments and at the very least this should be addressed by quantification assays for all these figures.

5) In Figure 5, the F220c fragment is a highly artificial construct consisting of a hybrid between the human and *c.elegans* TDP-43s. As such, its significance with regards to the real protein is rather questionable. Ideally, the authors should have complemented these experiments with a fragment that contains the TDP-1 region equivalent to human TDP-43 220-268 sequence. Does this construct behave in the same way as the F220c fragment?.

6) Another issue that is also lacking from the study is an analysis regarding the importance of the Delta 311-340 region in the full protein. It is not clear from the analyses whether this region is only important for the TDP-25 fragment (whose significance for the pathology is very much unclear as in many model systems it has not been detected) or whether it could have an importance for the full protein as well.

7) The proximity experiments reported in Figure 6 are quite interesting. However, they remain very descriptive, and it is not clear what could be the biological consequences that the C-terminal side of TDP-43 GRR, unlike its N-terminal side may have a steric hindrance. Have the authors performed toxicity studies on the different cellular populations to see whether this interaction can make a difference or not?

8) Finally, the *c.elegans* experiment in Figures 8-10 are an interesting approach but seem rather preliminary. In fact, it is rather disappointing that differences in locomotion at day 5 between the C220 and C233 are not highly significant (Figure 10A) and that survival seems not to be statistically affected (Figure 10B). This raises the question whether these truncations really make a big difference at the pathological level. Have the authors tried to see whether *c.elegans* stressed (for example, by raising the temperature or following pre-treatment with other stressors such as hyperbaric oxygen, juglone, or ionizing radiation) will show much more significant differences in the presence of C220 or C233?.

Reviewer #3 (Remarks to the Author):

The manuscript “Hetero-oligomerization of TDP-43 carboxy-terminal fragments with cellular proteins contributes to proteotoxicity” by Kitamura et al describes the different truncated versions of TDP-43 particularly at the RRM region at the N-terminus TDP-43 CTFs has intrinsically disordered regions which are able to form misfolded and hetero-oligomers. These truncated RRM regions also affected their propensity to condense/aggregate and lead to toxicity, which they demonstrate also in *C. elegans*. While

TDP-43 CTFs have been known to form condensates in the cell, this study was able to map out the region (220-262) of RRM2 as being potentially responsible for condensation formation. After reviewing this manuscript, I found the study extremely novel and interesting with a lot of take home messages that contribute to the field of TDP-43 dynamics and its effects on proteostasis dysfunction. I strongly feel that the study contributes greatly, particularly to the ALS research area. My comments regarding the paper are more about clarification of a few details.

- Have the different TDP-43 CTFs truncations been made recombinantly? If so, does the protein spontaneously form homo-oligomers? Could you comment on this? Or is the oligomerization usually only a consequence of when they are expressed in Neuro2a cells?

This would suggest that other proteins must be involved or influence the oligomerization (probably chaperones as referenced in the Discussion) may be contributing to the seeding event. Could the authors comment on the interactome or the proteome of the condensates? Do they suspect that these are the same proteins found in TDP-43 aggregates found in pathology? Or are they preceding events that seed aggregation?

- The different TDP-43 CTFs that are made expressed in Neuro2A or made recombinantly, have these been enriched to seed a newer batch of cells to see if they propagate and lead to aggregation of endogenous TDP-43? Does it behave as hypothesized as a prion?

- Given that the truncation affects the RRM region and NES of TDP43, do the different truncated forms of TDP-43 during condensation affect their distribution and cell location (nuclear/cytoplasmic) and have these been taken into consideration when calculating the % of condensates.

- What was a little unclear to me as I was reading was the sample numbers that were reported in the Figures and Figure Legends. E.g Figure 1C shows n=4, does this represent the number of times the experiment was carried out or the number of individual cells that were used for the analysis (n=3 or 4 seems really low). The percentages shown would indicate the former i.e. the entire experiment was carried out 4 times as biological replicates, and if so, what are the average numbers of cells counted per experiment to obtain the percentages shown? This is also the same with Figures 3-6 where a graph is showing "Cells containing condensates" n=3. C220 is showing 15% of cells with condensates from n=3. Could the authors clarify this either in the Legend or Methods.

Overall, I thought this was an elegant study which will greatly contribute to the understanding of TDP-43 dynamics.

Responses

To the Referee #1

In this work, Kitamura et al have investigated the effect of CTD fragments (CTFs) in the oligomerization of TDP-43. In his work, using both cellular and an animal (c.elegans) model they have investigated the influence of RRM2 misfolding on the amyloidogenic properties of the CTFs. In addition, the authors have taken advantage of their model systems to investigate the proximity properties of the N- and C-terminal regions of TDP-43. From the point of view of the novelty, it has been known since a long time that TDP-43 CTFs promote aggregation and are potentially toxic to cells, with many studies addressing this specific issue. Nonetheless, the work contains some novel data especially due to the use of advanced experimental techniques which have not been employed in the past and some interaction experiments (such as those reported for the C220 and C274 fragments) and that the N- and C-terminals TDP-43 CTFs are close to each other (although the hypothesis that TDP-43 N- and C-terminal interact with each other has already been proposed in the past by the Petrucelli lab several years ago). However, the work also contains several important weaknesses and, as a result, several specific and general issues will need to be addressed to improve the significance of the study:

1) First, the authors should provide some rationale for using the artificial truncation fragments shown in Figure 1A. The reason is that there is a lot of literature where cleavage sites in CTF fragments found in patients/enzymatic cleavage studies have been mapped precisely (ie. 174, 239, and 220/219 for the TDP25 fragment and 208, 247, 274, 291, 306 and 314 for lower size fragments. Would it not have been better to use all this information to plan for the CTF fragments to be used in this project? Second, in Figure 1 the authors should also add results for the C274 fragment for all experiments shown in this figure because this fragment is only used from Figure 2 onwards and is a key component for Figure 4 experiments.

[Response] Thank you for your insightful comments. Analyzing the aggregation propensity and toxicity of artificially truncated TDP-43 CTFs involves a straightforward and recursive approach using mutants with systematically depleted amino acids. Since Furukawa *et al.* reported that aggregation analysis of TDP-43 CTFs with every 5 residues removed from the N-terminus of C220 up to C245 (amino acid 245–414) showed that

they can be categorized into four distinct types [citation #51], we investigate the condensate formation of GFP-tagged TDP43 CTF variants adding N-terminal 10, 20, and 30 amino acid residues to C263 (C233, C243, and C253, respectively; shown in Figure 1A). First, regarding the rationale of the cleavage positions, we add a reference and an explanation for the TDP-43 fragmentation [DOI: 10.1016/j.isci.2021.102459] (Lines 6–11 on p. 4). The paper that analyzed aggregation by removing 5 residue increments from the N-terminus of C220 up to C245 was cited [DOI: 10.1016/j.bbadis.2011.09.005] and replaced with the description that CTFs with 10, 20, and 30 residues added to the N-terminus of C263 were analyzed (Lines 13–20 on p. 5). Second, we add results for the C274 fragment for all experiments shown in Figure 1.

2) Figure 2C is truncated and cannot be seen.

[Response] We apologize for the inadequate representation of this figure, which appeared to have occurred during the conversion to PDF on the submission system. This has been corrected.

3) Although they have been reported in Figure 1B, in Figure 3B the authors should also include images for C220 and C263 (in the current version only F220 RRM2 and SW are present). This is important because in the quantification of Figure 3C both C220 and C263 are present, and it is not clear if these are new experiments, or they refer to the data reported in Fig.1B.

[Response] The fluorescence images for C220 and C263 have been included in Figure 3D. Figures 3B & C are independent experiments, and we did not refer to the values represented in Figure 1. This explanation was added in Lines 25–27 on p. 7.

4) There is also an important discrepancy in the results for C220 in Figures 1D, Figures 3D/5E, and Figure 6F. In Figure 1D, the distribution of C220 in the S and P fraction is heavily predominant in the P fraction, in Figures 3D and 5E the distribution seems to be equal in the S and D fractions, and in Figure 6F the C220 fragment is present predominantly in the S fraction. All these huge differences raise the question of the variability observed in these experiments and at the very least this should be addressed by quantification assays for all these figures.

[Response] Thank you for pointing this out. The experiment has been performed multiple

times and we have added quantifications in Figures 1D, 3D, 5C, 6D, 7F, S5D, and S6C. These results were obtained in three independent trials of cell seeding, transfection, lysis, fractionation, and blotting. To normalize the blotting efficiency, the values are represented as insoluble (Insol.) proportions (see Material and Methods; Lines 19–21 on p. 24). For the blots in Figures 3D and 6D, C220, C263, and GFP monomers were included as controls. For Figure 7F, the represented blot (*left*) was modified so that the content of C220 in the pellet fraction was similar to Figures 1D, 3D, and 6D. Consequently, as we have shown the mean, error bars, and *p*-values obtained from the three independent trials in the revised one, the extent of the experimental errors can be estimated from the represented plots. We would like to emphasize that the quantification did not change the conclusion in this manuscript.

5) In Figure 5, the F220c fragment is a highly artificial construct consisting of a hybrid between the human and c.elegans TDP-43s. As such, its significance with regards to the real protein is rather questionable. Ideally, the authors should have complemented these experiments with a fragment that contains the TDP-1 region equivalent to human TDP-43 220-268 sequence. Does this construct behave in the same way as the F220c fragment?

[Response] According to this suggestion, we have tested a GFP-tagged fragment of TDP-1 (295-414 amino acids; TDP1-C295), corresponding to the human C220 sequence (220-414 amino acids). We analyzed its solubility, cytoplasmic condensation, and cell death (Supplemental Figure 5). Unlike F220c hybrid chimera, the condensation properties of TDP1-C295 were similar to C220. Consequently, TDP1-C295 did not behave in the same way as the F220c. However, these new results suggest that the combination of both tRRM2 (F220) and the C-terminal disordered region that are evolutionarily conserved may play an important role in the condensation of TDP-43 CTFs; thus, we modified a portion of the discussion part [Lines 25–35 on p. 17]. Since we believe that these results using F220c and TDP1-C295 are interesting and promising, we decided to keep these results in this manuscript but do not use them as a major basis for assuming the importance of TDP-43 GRR.

6) Another issue that is also lacking from the study is an analysis regarding the importance of the Delta 311-340 region in the full protein. It is not clear from the analyses whether this region is only important for the TDP-25 fragment (whose significance for the pathology is very much unclear as in many model systems it has

not been detected) or whether it could have an importance for the full protein as well.

[Response] Thank you for your insightful comments. Although cytoplasmic condensates/aggregates of TDP-43 CTFs such as C220 (TDP25) can be observed in Neuro2A and 293 cells, that of full-length TDP-43 cannot be observed even in its ALS/FTD-associated mutants nor the wild type in our system [DOI: 10.1038/srep19230; DOI: 10.3390/ijms24065513]. Our evaluation system for the cytoplasmic condensates of TDP-43 involves the expression of mutant TDP-43 carrying mutations in both NLS and 173/175th cysteines (TDP43CS) in 293 cells [DOI: 10.1101/2022.07.03.498631]. Therefore, we newly constructed a plasmid for expressing GFP-tagged TDP43CS lacking 311–340 amino acids region (HR) (TDP43CS Δ), and then we compare cytoplasmic condensation between TDP43CS and TDP43CS Δ using 293 cells (Supplemental Figure 6). Although the cells harboring cytoplasmic condensates of GFP monomers and wild type of TDP-43 as negative controls were not observed, those of TDP43CS and TDP43CS Δ were positive, but the difference between TDP43CS and TDP43CS Δ was very small (~1%; Supplemental Figure 6B). Next, to verify the slightly low efficiency of condensation of TDP43CS Δ biochemically, a filter retardation assay of cell lysates was performed as previously reported [DOI: 10.1101/2022.07.03.498631]. The amount of TDP-43 remaining on the cellulose acetate membrane with a 0.2 μ m pore size showed a slight but not significant decreasing trend with the lack of HR region (Supplemental Figure 6C). Although HR region may be involved in condensation of full-length TDP-43 more or less, it is unlikely dominant. A possible reason for this low contribution of HR region in the C-terminal GRR is likely that the N-terminal Ub-like domain in TDP-43 could be dominantly involved in the oligomerization and inclusion body formation [DOI: 10.15252/embj.2022111719; DOI: 10.1126/sciadv.adf6895]. However, the contribution of HR region may be more pronounced in TDP-43 CTFs likely due to the lack of the N-terminal Ub-like domains. Since this manuscript discusses the condensation mechanism of TDP-43 CTFs produced by cleavage or abnormal translation such as C220 (TDP25), we believe that it is important to show evidence of the contribution of HR.

7) The proximity experiments reported in Figure 6 are quite interesting. However, they remain very descriptive, and it is not clear what could be the biological consequences that the C-terminal side of TDP-43 GRR, unlike its N-terminal side may have a steric hindrance. Have the authors performed toxicity studies on the different cellular populations to see whether this interaction can make a difference or not?

[Response] To investigate the biological consequences of the N-terminal assembly of TDP-43 GRR, we first investigated whether amyloids were formed in the light-induced condensates; however, no cells harbored amyloid-positive condensates (Supplemental Figure 8A). Second, we investigated the viability of the cells expressing CRY-labeled C263 and C263 Δ using DRAQ7, a dead cell-specific staining dye, at 1 h after allowing blue light irradiation under the same conditions as in Fig. 7. However, there were no dead cells (Supplemental Figure 8B). Third, time-lapse microscopic observation showed that the condensates of CRY-C263 and -C263 Δ were disassembled by 30 min with a slight delay than that of CRY tag itself (Supplemental Figure 8C & D). Finally, the few C263 condensates that remained after 8 hours of blue light irradiation did not contain amyloids (Supplemental Figure 8E). These results indicate that blue light irradiation with a short time did not maintain the condensate formation of these GRRs. We next tested whether continuous exposure to blue light to maintain the condensate formation of C263 that causes cell death with ‘aging’ of the condensates. However, continuous blue-light irradiation for cells expressing CRY-labeled proteins caused cell death regardless of the expressed protein, suggesting that phototoxicity may be more dominant than cytotoxicity by GRR condensation. To investigate an appropriate condition exerting low phototoxicity, we have tried to change the power and interval excitation light irradiation; however, we have not established it yet. Therefore, we have not succeeded in clearing biological consequences of CRY-induced condensations. Although further studies are required, the experimental system using CRY may be to evaluate the propensity of proximity of GRR. Since light-induced condensates of full-length TDP-43 using CRY tag causes cytotoxicity [DOI: 10.1038/s41467-020-14815-x & DOI: 10.1016/j.neuron.2019.01.048], the N-terminal oligomerization domain of TDP-43 would be involved in toxic conformational changes of TDP-43 with condensations. These considerations are added from line 18 on p. 13 to line 4 on p. 14.

8) Finally, the c-elegans experiment in Figures 8-10 are an interesting approach but seem rather preliminary. In fact, it is rather disappointing that differences in locomotion at day 5 between the C220 and C233 are not highly significant (Figure 10A) and that survival seems not to be statistically affected (Figure 10B). This raises the question whether these truncations really make a big difference at the pathological level. Have the authors tried to see whether c.elegans stressed (for example, by raising the temperature or following pre-treatment with other stressors such as hyperbaric oxygen, juglone, or ionizing radiation) will show much more significant differences in the presence of C220 or C233?

[Response] According to this suggestion, we have performed lifespan assays under stressed conditions. We tested worm survival during juglone treatment. On the NGM plate containing 270 μ M juglone, synchronized worms (day 1 adulthood) expressing G-C220 were rapidly dead than those expressing G-C233 and GFP monomers (Median life expectancy 4 h for C220, 6 h for C233, and >8 h for GFP monomers; Figure 9H). This condition clearly showed a difference in toxicity among C220, C233, and GFP. Therefore, C220 was the most toxic during the mild oxidative stress compared to C233 and GFP monomers, and C220 was more toxic than C233.

To the Referee #3

After reviewing this manuscript, I found the study extremely novel and interesting with a lot of take home messages that contribute to the field of TDP-43 dynamics and its effects on proteostasis dysfunction. I strongly feel that the study contributes greatly, particularly to the ALS research area.

1) Have the different TDP-43 CTFs truncations been made recombinantly? If so, does the protein spontaneously form homo-oligomers? Could you comment on this? Or is the oligomerization usually only a consequence of when they are expressed in Neuro2a cells? This would suggest that other proteins must be involved or influence the oligomerization (probably chaperones as referenced in the Discussion) may be contributing to the seeding event. Could the authors comment on the interactome or the proteome of the condensates? Do they suspect that these are the same proteins found in TDP-43 aggregates found in pathology? Or are they preceding events that seed aggregation?

[Response] We have tried to express two TDP-43 CTFs (C220 and C233) in *E. coli* recombinantly; however, the yield has been extremely low and we were unable to recover soluble protein. Therefore, we cannot comment on the oligomerization status of purified C220 and C322 proteins at this time.

C220 and C233 are partially soluble in Neuro2a, which could be related to expression levels or the presence of other factors that help to stabilize TDP-43 CTFs inside cells. We have determined cytoplasmic HSP70 as an interacting protein and an aggregation-suppressor of GFP-C220 in Neuro2a cells [DOI: 10.1111/gtc.12495 and

10.1007/s12192-018-0930-1; citations #68 and #69 in this manuscript, respectively]. Furthermore, soluble GFP-C220 from Neuro2a cell lysate was able to be purified with HSP70 [DOI: 10.1101/2022.07.03.498631; citation #58 in this manuscript]. These findings suggest that HSP70 may suppress C220 aggregation while maintaining its solubility. On the other hand, interactome analysis with TDP-43 aggregates showed that HSP70 recruited the aggregates, supposing that TDP-43-mediated perturbation of the HSP70-associated signaling pathway, which may contribute to cellular dysfunction associated with TDP-43 pathology [DOI: 10.1016/j.isci.2023.106645]. Regarding the preceding events of HSP70 to seed aggregation, HSP70 and HSP110 machinery work as disaggregases of various aggregation-prone proteins, and enhanced disaggregation might generate short fibrils and oligomeric species from amyloid aggregates, thereby eventually increasing amyloid seeding capacity and enhancing prion-like spreading of disease conformers. [DOI: 10.1016/j.molcel.2018.01.004]. As a result, HSP70 may be responsible for a balance between inhibiting the aggregation and the disaggregation of TDP-43 CTFs. This discussion was newly added in Discussion part of our revised manuscript [from the last line on p. 18 to line 7 on p. 19].

2) The different TDP-43 CTFs that are made expressed in Neuro2A or made recombinantly, have these been enriched to seed a newer batch of cells to see if they propagate and lead to aggregation of endogenous TDP-43? Does it behave as hypothesized as a prion?

[Response] As mentioned in Q1, the production of recombinant TDP-43 CTF protein is technically very challenging. To test whether TDP-43 CTFs propagate and lead to aggregation of TDP-43 as a prion, we have tried to measure co-aggregation between exogenous human TDP-43 CTFs and endogenous murine TDP-43 using antibodies for immunofluorescence. Due to the lack of available antibodies that recognize only mouse epitopes of TDP-43, we have utilized co-aggregation of TDP-43 with its CTFs was thereby analyzed using co-expression of GFP-tagged TDP-43 CTFs (G-C220 and G-C233) and RFP-tagged TDP-43 (TDP43-R) (Figure 5). TDP43-R was colocalized with the condensates of G-C220/C233 (Figure 5, A & B). G-C220/C233 expression increased the amount of insoluble R-TDP43FL (Figure 5C). These results suggest that C220/C233 may behave like a prion against full-length TDP-43. These were newly added in our revised manuscript [Lines 13–29 on p. 9 and from line 30 on p. 11 to line 17 on p. 12].

3) Given that the truncation affects the RRM region and NES of TDP43, do the

different truncated forms of TDP-43 during condensation affect their distribution and cell location (nuclear/cytoplasmic) and have these been taken into consideration when calculating the % of condensates.

[Response] Although nuclear import of TDP-43 is active through its NLS sequence, its nuclear export is passive, and NES sequence (239–250 amino acids) is not solvent-exposed [DOI: 10.1038/s41598-018-25008-4, 10.1038/s41598-018-25007-5, etc.]. We think that it is important to evaluate whether NES in TDP-43 CTFs affects the cytoplasmic condensation, thereby, we used GFP-tagged C220 carrying NES mutations (I239A, L243A, L248A, I249A, and I250A; G-C220mNES). First, fluorescence intensity in the nucleoplasm was slightly lower than that in the cytoplasm both G-C220 and G-C220mNES (Supplemental Figure 1). Second, no significant difference in fluorescence intensity of nucleoplasm relative to cytoplasm except for the condensate region between G-C220 and G-C220mNES in Neuro2a cells was observed (Supplemental Figure 1B). Furthermore, the condensation efficiency of G-C220mNES was not changed to that of G-C220 (Supplemental Figure 1C). Therefore, NES in RRM2 region did not affect the condensation efficiency of C220, and we conclude the intracellular distribution of TDP-43 CTFs was not involved in their cytoplasmic condensation (from line 29 on p. 5 to line 8 on p. 6) .

4) What was a little unclear to me as I was reading was the sample numbers that were reported in the Figures and Figure Legends. E.g Figure 1C shows n=4, does this represent the number of times the experiment was carried out or the number of individual cells that were used for the analysis (n=3 or 4 seems really low). The percentages shown would indicate the former i.e. the entire experiment was carried out 4 times as biological replicates, and if so, what are the average numbers of cells counted per experiment to obtain the percentages shown? This is also the same with Figures 3-6 where a graph is showing “Cells containing condensates” n=3. C220 is showing 15% of cells with condensates from n=3. Could the authors clarify this either in the Legend or Methods.

[Response] The ‘n’ refers to the number of independent experiments performed. We have modified the figure legends to indicate the number of independent trials and added the total number of cells that were quantified.

Overall, I thought this was an elegant study which will greatly contribute to the

understanding of TDP-43 dynamics.

REVIEWERS' COMMENTS:

Reviewer #1 (Remarks to the Author):

Authors have performed several experiment to substantiate their claims and have correctly acknowledged some limitations of their study. As a result, I think they should be commended and there are no further objections from this reviewer.

Reviewer #3 (Remarks to the Author):

The authors have carried out additional experiments and addressed my specific comments and feedback.